# Sequence structure in children's speech reveals non-linear development of relations between word categories
Maja Linke [1] ✉ & Michael Ramscar [2]

Why do children learn some words earlier than others? Can children's speech patterns reveal how their evolving models of language determine what they learn? This study presents a systemic analysis of children's speech using low-dimensional embeddings to examine how the contextual knowledge reflected in their utterances reorganizes as linguistic experience increases. We analyzed age-stratified samples from the CHILDES database (18–36 months: $n = 1,693,641$ tokens; 3–6 years: $n = 1,750,007$; 5–12 years: $n = 1,721,828$) and adult speech from the SUBS2VEC subtitle corpus ($n = 1,742,885$). Our results suggest that the order and position of words in sequences produced by children from different age groups reflect changes in the way they represent categories of words. Rather than being ungrammatical, children's utterances appear to be structured by temporary grammars that optimize the distribution of information in sequences. The results point to shifts in how words are organized in semantic space, reflecting the gradual alignment of lexical categories during learning; this restructuring appears to draw on functionally ambiguous (multipurpose) categories in English. These findings are somewhat counterintuitive, as they suggest that not knowing the exact meaning of words can facilitate both learning and communication.

Speech offers an interface to the mind. What people talk about and how they say things can reveal what they notice, remember, and attend to. These tendencies are shaped by individuals' experiences and their ability to align their present context with their priors. In this study, we focus on children's speech, which often includes incomplete or ungrammatical elements. We analyze the patterns in children's productions to examine whether they can offer insight into the way that children's first models of language develop. These temporary representations appear to help children master aspects of language that are difficult to grasp later in life. While the time it takes children to develop adult-like patterns varies, this learning process typically follows a U-shaped progression. Initially, children may use grammatical, adult-like patterns but then seem to lose this ability before regaining it later on. Such nonmonotonic learning trajectories occur across various modalities and at different levels of abstraction at different points in development[1,2]. By performing systemic analyses of children's early utterances, we seek to illuminate the network dynamics that give rise to these patterns in language learning as they unfold over time. Accordingly, rather than focusing on learning trajectories, this approach aims to identify the distributional factors that influence the development of early mental models

and offer new perspectives on conditions that determine how learning evolves.

It has long been known that children learn different word categories at different rates and that their grasp of how to use words follows a similar pattern. Nouns, for example, are generally learned before verbs, and verbs are more challenging for children to learn and recall[3–5]. Children often know color names before they can match them with the correct colors[6–8], and they learn number words before grasping their actual magnitudes[9,10]. Similarly, young children use time-related words like *tomorrow* or *last week* before they can accurately place them on a timeline. A 3-year-old may talk about these time points, but struggle to order them correctly on a line, and understand that *yesterday* and *last week* are closer together than *last week* and *last year* take years to develop[11,12]. That is, it is clear that some word categories may play a role in sentence structure even before they are fully understood. It follows that children must, at least in part, learn about language structure without having to establish adult-like connections between words and real-world events. Similarly, knowledge of the relationships between words and how they fit into sentences must, at times, develop separately from understanding how those words relate to actual events. This

[1]Max Planck Institute for Human Cognitive and Brain Sciences, Leipzig, Germany. [2]Department of Psychology, University of Tuebingen, Tuebingen, Germany. ✉e-mail: linke.maja@gmail.com

separation can be convenient because it allows children to communicate effectively even when their experience with the world and words is incomplete. Over time, communication helps them align their understanding of language with real-world experiences, even when those experiences are unique to each individual and do not have straightforward linguistic labels[13].

It is clear that language users utilize context to interpret words and that their interpretations vary according to individuals' knowledge of the structures that represent these contexts[14–16], along with their certainty about the structures[17–19]. It is also clear that children learn many aspects of the meanings of words from the linguistic context in which they appear [see, e.g., ref. 20]; However, how children use context to learn words (and other communicative signs) remains relatively unexplored. Most research on word learning has focused on individual words and categories of words without considering the statistical properties of the contexts in which these words appear. Yet, communication patterns are complex and evolve over time, both across individuals and societies[21]. Isolated examples taken out of context are bound to miss the broader network dynamics at play. By analyzing developmental patterns at the level of systems as a whole and relating them to the characteristics of their parts, such as individual words or short phrases (n-grams), we can reveal insights that are otherwise obscured.

Recent generations of language models (LLMs) demonstrate how patterns in word co-occurrence and, importantly, segmentation can provide sufficient information to capture many aspects of human cognitive abilities [or at least mimic; cf. 22–24]. These capabilities of the models, in turn, provide a motive for revisiting the question of how children gradually acquire language by learning from the natural distribution of words and other features/signs in speech [see e.g., refs. 25–27] and how experience influences the informativeness and function of these features/signs[28,29]. The meaning and function of communicative contrasts—phrases, words, and sounds—are shaped by their relationships with other parts of the language system. This implies that the state of the system represented by the individual learners codetermines how speakers interpret messages and how useful/informative individual words are in communication. From this perspective, learners' knowledge about the structure of the language system itself defines and constrains the scope and specificity of word meanings.

In what follows, we analyze how language development influences systems of relations (distributed representations) and how changes in these systems are reflected in speech production patterns. Prior work has analyzed child-directed speech with high-dimensional models to illuminate children's learning mechanisms[14,30]. By contrast, instead of modeling how children learn, we use low-dimensional models of children's productions to uncover what they learn and when, aiming to clarify the nature of their developing linguistic representations. To this end, we trained low-dimensional semantic networks on transcripts of word sequences produced by adults and children of different ages. By tracking the relationships between categories of words that differ in their distributional properties and the age at which children typically master them, we estimated changes in the underlying structure. We use these word categories as developmental markers to guide our analysis.

In a multipart analysis, we address the following questions:

(1) Do word embeddings trained on data from younger cohorts better discriminate between word categories learned earlier (e.g., nouns) than embeddings trained on older speakers' data? If errors or differences in children's speech are arbitrary deviations from adult language models, these differences should be uniformly distributed across word categories. Suppose, however, error patterns change with a child's growing knowledge of the language system and the distribution of uncertainty associated with different categories. In that case, the differences between child and adult models should be evident in the category embeddings. Do production patterns differ systematically across age groups?

(2) Is there a qualitative change in the structure of the latent semantic space? The rate at which children learn words from different categories varies significantly in the first few years of life. Can the order in which

words from these categories are learned be inferred from the structure of semantic representations, and do these structures develop to support discrimination between categories?

(3) Are deviations in younger cohorts' production patterns consistent with model differences? Are the differences in clustering patterns aligned with changes in word positions within sequences and fluctuations in the probabilities of words from different categories?

## Methods
### Data
**Corpus data.** The speech transcripts used in this study were sourced from two primary databases: the CHILDES (Child Language Data Exchange System) database for child-caregiver speech transcripts accessed through the `childes-db` [interface[31–33], version 2021.1], and SUBS2VEC, a publicly available subtitle corpus for adult speech transcripts[34]. We grouped the CHILDES data by age: 18–36 months ($n = 1{,}693{,}641$ tokens), 3–6 years ($n = 1{,}750{,}007$), and 5–12 years ($n = 1{,}721{,}828$). These cohort boundaries were set to track major shifts in cognitive development and literacy learning. For the adult sample ($n = 1{,}742{,}885$ tokens), we used subtitle data from various English-language television shows and movies. A detailed list of collections used in these analyses is available in SI). For model training, we used cohort-balanced subsets to ensure comparable exposure across age groups; however, all statistical analyses were also conducted on the full, unbalanced dataset (TO = 3,177,782; 3PLUS = 2,947,716; 5PLUS = 1,716,566; ADULTS = 2,929,149 tokens), and all results replicated.

**Behavioral data.** We assessed how well the embeddings predict children's performance in semantic discrimination and sound discrimination using behavioral responses (accuracy and reaction times) from a publicly available longitudinal dataset, *Language Processing in Children* [OpenNeuro `ds003604`[35]]. The dataset includes semantic and phonological discrimination trial-level responses from 322 monolingual English-speaking children, collected longitudinally at ages 5 ($N = 139$; $M_{age} = 68.7$ mo, SD = 3.2 mo), 7 ($N = 280$; $M_{age} = 89.1$ mo, SD = 4.1 mo), and 9 years ($N = 101$; $M_{age} = 110.4$ mo, SD = 2.1 mo).

In each trial, children listened to word pairs and judged whether they were semantically related (semantic discrimination) or whether they sounded similar (sound discrimination). Semantic pairs ranged across low-association (e.g., *water-drink*), high-association (e.g., *syrup-pancake*), and unrelated (*flush-cliff*) conditions. Sound-discrimination pairs included rhyme (e.g., *wide-ride*), initial-sound (e.g., *coat-cup*), and unrelated (*zip-cone*) conditions. We extracted trial-level reaction times and binary accuracy scores.

**Ethics and data use.** This study analyzes publicly available, fully de-identified datasets (CHILDES; OpenNeuro DS000221). No new data were collected. All ethical approvals and informed consent procedures were obtained by the original investigators and are documented in the publications describing these datasets [e.g., refs. 31,35]. Our analyses comply with each repository's terms of use, and no additional institutional review was required for secondary analysis of de-identified data.

**Demographic information.** For the OpenNeuro dataset (DS000221), age and sex information were provided by the original investigators. No additional demographic variables were available. For CHILDES, only child age is consistently reported across contributing studies, and analyses were grouped by age alone.

**Preregistration.** This study was not preregistered.

### Embedding models
**Training procedure.** We trained shallow, 15-dimensional word2vec models separately for each of the four cohort samples to learn vector representations of words. We used the R implementation of

word2vec[36,37]. The transcripts and subtitles were lowercased, tokenized, and part-of-speech tagged with spaCy using the `en_core_web_sm` model[38]. We experimented with embeddings of 10, 15, and 20 dimensions; because systematic differences were not evident, the dimensionality was set to 15. The context window size was 5 (i.e., each word vector was influenced by the five preceding and following tokens). Training parameters were held constant across cohorts (Epochs: 20; Negative Sampling: 5; Learning Rate: 0.01).

**Model rationale**. Rather than treating word2vec as a model of learning, we use the 15-dimensional embedding space as a static baseline (i.e., a fixed probe model), a constant reference geometry that functions like an intercept in regression, letting us quantify how speech structure evolves across timescales and speakers without claiming that the embedding itself exhaustively captures that structure. Because training relies on aggregated co-occurrence statistics, the model is cohort-neutral and fixed in time, locked to a single grain of representation: whole-word segmentation.

Our approach is motivated by the observation that variability in segmentation appears to be a key property of sparse intelligent communication systems, where domain-specific information relies on specialized communication patterns[39,40]. Since word boundaries are products of evolved writing systems, the extent to which speakers adhere to these word or segment boundaries in speech is likely to vary depending on their familiarity with the corresponding writing system and the variation in the information structure of its subsamples (i.e., text tailored to expert audiences vs. text tailored to general audiences).

Thus, we diagnose development by quantifying divergence from this fixed geometry rather than by fitting a trajectory. We interpret departures not as model error but in terms of multi-timescale restructuring driven by evolving attention and representational capacity. This approach differs from factorization methods such as topic modeling[41,42] or more conventional approaches to latent semantic analysis[14,16]: The information lost in the shallow network is conceptually important. Our goal is to establish a coarse, low-resolution baseline structure that serves as a reference against which developmental restructuring can be measured.

**Comparison to other architectures**. Non-embedding baselines used in sections "Learning redistributes and smooths information in sequences" and "Developmental changes are most prominent at sequence boundaries" model raw distributions directly (frequencies, position-wise categories, and short *n*-grams), whereas embedding models compress the same local co-occurrence into a low-dimensional similarity space that can serve as a single predictor. We used word2vec because it is efficient, relatively interpretable, and deliberately context-insensitive; by encoding only local co-occurrence within a fixed window, it provides a stable yardstick for cohort comparisons. Alternatives differ in ways that are not aligned with this goal: GloVe factorizes global co-occurrence and is more sensitive to corpus-wide frequency structure and hyperparameters; BERT-type models yield contextual token-level embeddings that typically require additional fine-tuning and introduce greater computational complexity. In practice, these alternatives can blur the sequence-level shifts we seek to isolate.

**Evaluation and exploratory contrasts**. We used the word2vec representations of words from preselected target classes to examine shifts in the relations between syntactic and semantic categories. To address our research question, we analyzed subsets of words that satisfy three criteria. First, we included closed-class vs. open-class contrasts (e.g., kinship terms vs. names). Second, we examined words that share open-class properties but represent distinctions along the abstract-concrete continuum (e.g., food words, names, time words). Third, we considered part-of-speech labels representing grammatical vs. semantic contrasts, with pronouns and nouns as the extremes of the functional continuum (semantic heuristics vs. embedded contrasts) and adverbs and adjectives as intermediate (embedded modifiers).

To assess sensitivity to training order and parameter variation, we compared CBOW and skipgram embeddings while manipulating dimensionality, window size, and utterance structure. We projected embeddings into two dimensions with *t*-SNE to examine whether categories formed coherent structures across models.

In addition, we trained 15-dimensional CBOW models on the same age-stratified samples under four sequence manipulations: (1) *forward* (original order), (2) *backwards* (utterances reversed), (3) *shuffled* (tokens randomly permuted within utterances), and (4) *position-coded* (utterances augmented with position tokens). Exploratory results are provided in the supplement and a companion notebook with robustness checks is available in the OSF repository.

### Structural analyses of embedding spaces

**t-SNE analyses**. We visualized the geometry of a two-dimensional representation space/spatial distribution of categories for each cohort using distributed stochastic neighbor embedding (t-SNE). This nonlinear dimensionality reduction algorithm maps high-dimensional data points into a lower-dimensional space (in this case, a two-dimensional map). tSNE is an embedding technique that maps data points that are close together (similar) in the higher-dimensional space so that they stay close together in the low-dimensional map. It does this by computing joint probability over pairs of data points and fit points in a low-dimensional map that maintains minimal Kullback-Leibler divergence between the probability distributions representing the higher and lower dimensional space. The perplexity (optimal number of neighbors) was held constant at 20. First, we visually inspected the two-dimensional maps to understand how category relationships change across developmental stages. In addition, we performed Procrustes analyses on the t-SNE coordinates and further statistical tests on the matrices (Mantel test and Spearman correlation of the upper triangular part of the matrix) and individual category Euclidean and Mahalanobis distance scores.

**Procrustes analyses**. We conducted Procrustes analyses to estimate changes in the shape of the latent space based on embeddings, t-SNE coordinates, and distance matrices. These shape analyses allow us to examine the alignment between data points from different cohort models, representing words from various categories. The analysis evaluates the degree of rotation needed to align two models' data points. It determines how much additional translation and scaling are required to match individual data points best. By inspecting the residual error, we can identify groups of data points (categories) with the largest differences between models.

Procrustes analysis is a common algorithm used to align data points in multivariate spaces, where identical data sets are collected at different sites or at different points in time by applying scaling, rotation, and translation. We used vegan for R implementation of the Procrustes rotation and Procrustes permutation tests to estimate the significance of the test statistic. For distance metrics, we tested Euclidean and Mahalanobis distances.

**Distance metrics**. We compared Cosine, Euclidean and Mahalanobis distances of individual clusters. Cosine distance is a measure of the difference between two vectors, calculated as one minus the cosine of the angle between them (values range from 0, indicating identical, to 2, indicating opposite). While cosine distance is commonly used in text analysis and information retrieval to assess how similar two documents are, Mahalanobis distance is a more general measure that takes into account the correlations between variables and scales them according to their variability. It is used to determine the distance between a point and a distribution, considering the shape of the data distribution.

We find no noteworthy differences between the distance metrics.

**Mantel tests**. To analyze differences between the cohort embeddings, we employed the Mantel test, a statistical method used to assess the correlation between two distance matrices. We used the implementation of the

Mantel test provided by the vegan library for R[43], which supports permutation-based significance testing. Specifically, pairwise distance matrices were calculated for the embeddings, and 9999 permutations (permuting rows and columns of the first dissimilarity matrix) were performed to assess statistical significance.

### Sequence-level analyses and statistical modeling

**Lexical diversity GAMM.** To model the relationship between lexical diversity, utterance position, and speaker age, we used a generalized additive mixed model (GAMM) implemented in the mgcv package for R[44]. A GAMM is an extension of generalized additive models, allowing for the inclusion of random effects to account for non-independence in hierarchical or grouped data. It flexibly models non-linear relationships between predictors and the response variable using smooth functions. The formula used for the model was: $\text{LexicalDiversity } s(\text{UtterancePosition})+ \text{Cohort} * \text{POS} + s(\text{UtterancePosition}, by = \text{Cohort}, bs =' fs', m = 1)$. In this model, $s(\text{UtterancePosition})$ captures the overall smooth trend of lexical diversity across token positions, while the interaction of Cohort and POS accounts for the combined effect of speaker age and part-of-speech category. The term $s(\text{UtterancePosition}, by = \text{Cohort}, bs =' fs', m = 1)$ applies a factor-smooth interaction, allowing for cohort-specific smooths of token position using a functional spline basis with a penalty ($m = 1$) for smoothness. This setup accounts for non-linear trends and variability across cohorts.

We truncated utterances at position 25, removing 0.26% of tokens (28,502 words), because positions beyond this point were too sparse and inconsistently represented across cohorts. Because utterance positions are positive and strongly right-skewed (approximately exponential), we used a Gamma family with a log link to model position effects.

**Information-theoretic measures.** To examine redistribution of uncertainty within sequences, we computed adjacent-category mutual information (MI) and Jensen-Shannon divergence (JSD). MI quantifies how much knowing the current part of speech reduces uncertainty about the next part of speech. JSD measures the divergence between the position-specific distribution of categories and the global baseline distribution (first-order JSD) or between position-specific transition probabilities and their baseline (second-order JSD). Both measures capture position-sensitive developmental differences.

**Statistical analysis and reporting.** Analyses combined non-parametric and semi-parametric models implemented in R. Mantel tests were used to assess correlations between distance matrices; as permutation-based procedures, they do not yield degrees of freedom or analytical confidence intervals. We therefore report the Mantel statistic ($R$), permutation-based $p$ values, and the number of permutations.

Developmental effects were modeled using generalized additive mixed models (mgcv). Smooth terms were estimated with penalized thin-plate regression splines. For each smooth, we report the effective degrees of freedom (edf), approximate $F$-values, and two-sided $p$ values. Uncertainty around smooth functions is visualized using the approximate Bayesian credible intervals returned by mgcv. These intervals express uncertainty in the estimated smooth functions (i.e., the shape of the fitted curves), rather than confidence intervals for individual parameters. In contrast, the parametric components of the model (e.g., cohort and category contrasts, linear effects) are unpenalized and behave like standard linear-model coefficients. For these terms, we report standard errors and corresponding 95% Wald confidence intervals. Model diagnostics (gam.check) indicated that the basis dimension $k$ was sufficient and that residual structure and model fit were acceptable.

**Assumption checks.** Normality and homogeneity of variance were not formally tested because the procedures used here (GAMMs, permutation-based Mantel tests, information-theoretic measures, and

Procrustes analyses) do not rely on parametric assumptions about the distribution or variance structure of the underlying data.

The developmental data and systemic assumptions we are working with present challenges for applying classical statistical tests. Our analyses summarize trends and quantify relationships, but are not used to support claims that depend on classical parametric inference. We hypothesize that the relationship between sequence positions and the subjective experience of time (specifically, attention to segmental and word boundaries in continuous time) changes with learning. This kind of relational change inherently violates many of the assumptions of traditional statistical models, which typically assume stable, independent, and linear relationships between variables.

This limitation is neither new[45,46], nor unique to the present work: it applies to all analyses involving naturalistic behavioral data and language data, especially in developmental contexts. Complex systems, such as human communicative systems, exhibit distributional and temporal properties that are difficult to reconcile with the assumptions underlying mathematical models. We believe this is an important consideration for future research in this area.

### Reporting summary

Further information on research design is available in the Nature Portfolio Reporting Summary linked to this article.

## Results

### Between-category discrimination precedes within-category discrimination

To test whether the models of the younger cohorts aligned with the patterns seen in early learning, we examined whether developmental differences in production affect how models separate categories from one another (as opposed to distinguishing words within the same category) (Fig. 1). We compared category level distances across cohorts using Euclidean and Mahalanobis measures to assess whether models trained on the different age groups transcripts diverge systematically in their relational structure.

We began by visually exploring category differences through correlogram maps of distance values, comparing these across cohorts (Fig. 2b). We also compared cohort models by subtracting matrices from one another and visualizing the difference matrices (Fig. 3c). These analyses revealed that differences between categories increase with age in children but decrease in adults. In contrast, within-category differences decrease in children and increase in adults.

Next, we correlated matrix values across age cohorts, interpreting strong correlations as evidence of stability and weak correlations as evidence of developmental change. Mantel tests confirmed significant relationships between all cohort model pairs ($R > 0.509$, $p = 0.001$, 9999 permutations). We also tested for the relationship between individual categories. No statistically significant associations were found for names ($R = 0.142$, $p = 0.304$) or color words ($R = 0.387$, $p = 0.130$; Mantel test, 9999 permutations), and the strength of relationships decreased between high-frequency common nouns and closed-class words (see Supplementary Table S2). In contrast, stable relationships were maintained for verbs ($R = 0.581$, $p < 0.001$) and time words ($R = 0.476$, $p < 0.001$). Overall, matrix similarity decreased with greater age differences (i.e., toddler and 3–6-year-old models were more similar than 5–12-year-old and adult models), indicating that cohort-specific language models captured similar relational structures that changed progressively over development. The degree to which subsets of the matrix were affected by this change varied across categories.

The biggest differences between models were observable in the contrasts between names and common nouns (food words) and between numbers and kinship terms. Differences also emerged between pronouns and abstract terms (represented by time words), which are stronger than the differences between verbs and pronouns. This pattern of results indicates that changes in the relations between word categories have different effects

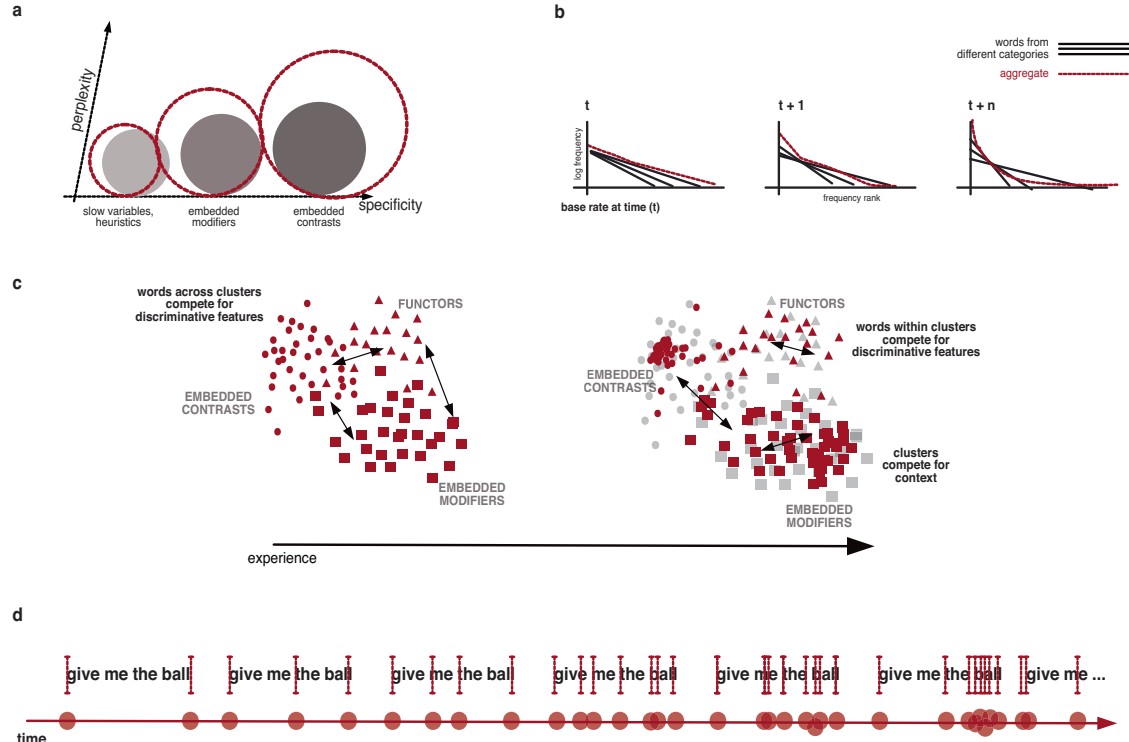

**Fig. 1 | The dynamics of learning from sparse distributions.** We hypothesize that learning is an adaptive response that maintains the discriminability of syntactic and semantic clusters. **a** Staggered development of three exemplary categories: functors, embedded modifiers, and embedded contrasts; because of the way words are distributed in language, the number of new words that children learn over time ought not to be evenly distributed across categories. Moreover, the function of these words in sequences and the uncertainty associated with the transitional probabilities between words from different categories ought to change over time. **b** Consequences of staggered learning on the distribution of words from the exemplary categories (straight lines) and their impact on the shape of the aggregate distribution (red dashed line). **c** Hypothesized change in geometry of latent representations, visualized in two-dimensional space, and **d** corresponding changes in the temporal structure of sequences with experience. The red lines represent event boundaries.

on the contrasts within the cluster in the open and closed categories, depending on the grammatical function of the words (Fig. 3c).

## The development of category differences is U-shaped

Across child cohorts, the distance between noun clusters and other clusters increased, whereas in the model trained on adult data, this distance decreases (see Supplementary Fig. S12). At the same time, the average distance between individual noun items becomes smaller, indicating that the within-cluster contrast in children's speech increases. Conversely, this within-cluster discrimination decreases in the models trained on adult speech, resulting in a U-shaped development of the semantic distance between and within noun clusters. We also observed differences in the mean distance of categories, which gauges the uncertainty associated with features of individual categories. The contrast between categories increased steadily in children, whereas discrimination within the category decreased: food nouns became less similar to other food nouns and more similar to all other nouns (Fig. 3c). The relationships between the models were preserved, but the correlations weakened with increasing speaker experience.

In contrast, strong relationships were preserved between clusters of verbs ($n = 30$; Mantel statistic: TO-3PLUS $R = 0.718$, $p < 0.001$; 3PLUS-5PLUS $R = 0.765$, $p = 0.002$; 5PLUS-ADULT $R = 0.581$, $p < 0.001$), pronouns ($n = 7$; TO-3PLUS $R = 0.884$, $p < 0.001$; 3PLUS-5PLUS $R = 0.854$, $p < 0.001$; 5PLUS-ADULT $R = 0.710$, $p < 0.001$), and numbers ($n = 12$; TO-3PLUS $R = 0.916$, $p < 0.001$; 3PLUS-5PLUS $R = 0.890$, $p < 0.001$; 5PLUS-ADULT $R = 0.700$, $p = 0.026$). These results show that the degree to which individual categories are affected by global redistribution varies locally across the lexicon. As more productive clusters, such as nouns, are updated, the

features that distinguish between lexical categories also shift, altering their contribution to sequential organization.

## Qualitative differences in cluster geometry are correlated with the perplexity of individual categories

To address our second research question more directly, we examined whether developmental shifts produce qualitative changes in the geometry of the latent semantic space. Two-dimensional t-SNE maps showed differences in cluster density, distances between clusters, and distances between clusters and the centroid.

We quantified these differences by performing Procrustes analyses on raw embeddings, t-SNE coordinates, and distance matrices. Permutation tests based on correlations from symmetric Procrustes analyses supported coherence between the cohort models (all permutation tests $p < 0.001$). For comparisons involving the toddler model, correlations after rotation ranged from 0.518 to 0.669 (TO-3PLUS: $R = 0.669$, SS = 0.553; TO-5PLUS: $R = 0.592$, SS = 0.650; TO-ADULT: $R = 0.518$, SS = 0.732). Comparisons among the older cohorts yielded correlations between 0.569 and 0.639 (3PLUS-5PLUS: $R = 0.639$, SS = 0.592; 3PLUS-ADULT: $R = 0.569$, SS = 0.676; 5PLUS-ADULT: $R = 0.587$, SS = 0.656). Detailed category-level correlations are provided in Supplementary Table S3.

In this context, coherence indicates that the low-dimensional model captures the relevant features of the representation space and identifies the empirical clustering relations, despite differences between the models (the residual sum of squares, which lies between 0.553–0.732, represents the misfit between the shapes, showing greater shape dispersion between children's and adults' matrices). The Procrustes error analysis revealed differences in scale and translation within noun clusters, whereas verb clusters

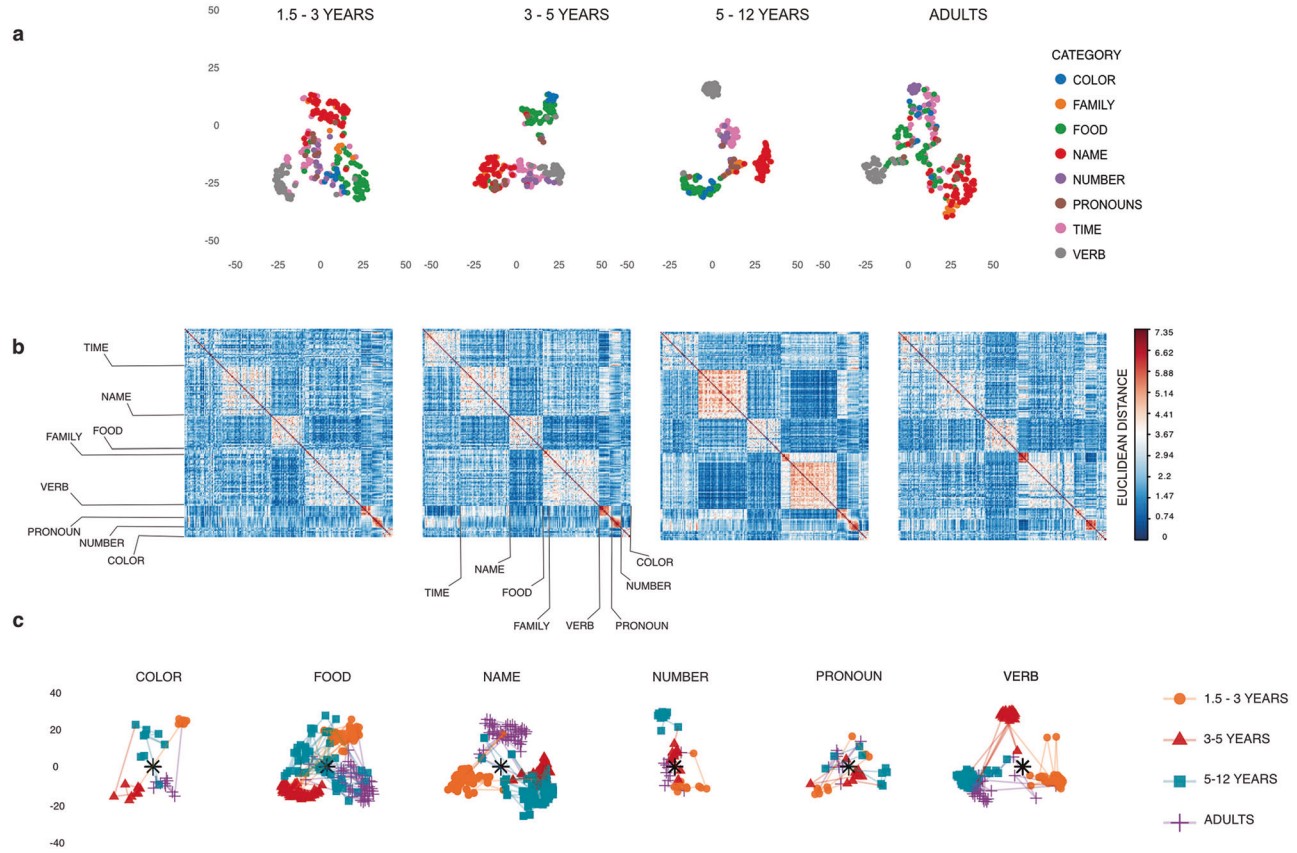

**Fig. 2 | Word2vec models trained on different cohorts' speech transcripts.**
**a** Differences between cohorts' representational geometries visualized with Sto-chastic Neighbor Embeddings (t-SNE). As predicted, cluster density and distance from the centroid increase more in the productive classes (verbs and nouns) than in closed classes (pronouns). The low-dimensional networks trained on adult speech are more similar to toddler networks than to older children's networks. **b** Semantic distance between word vectors from target categories. The emerging rectangular pattern shows the increase in differences between the categories. The decrease in the differences between words within categories is visible in deeper, more consistent shading of the rectangles lined up against the diagonal. The contrast between clusters increases with age in children and decreases in adults, leading to a U-shaped development of the low-dimensional geometry. **c** Differences between cohorts' embeddings for six target categories visualized with t-SNE. Distances between points approximate distances between word vectors; cluster density and distance from the centroid index cohort divergence.; the full set of eight categories is shown in Supplementary Fig. S13.

shifted coherently across the feature space. That is, transformations between cohorts' verb embeddings are linear. Over time (represented by the age differences among cohorts), verb clusters migrated together, maintaining their internal structure while shifting their overall position in response to evolving production patterns in the more productive word categories.

**Word embeddings predict behavioral responses in word similarity and sound discrimination task**

We evaluated how well cohort-trained CBOW embeddings captured variation in children's behavioral responses in the longitudinal OpenNeuro dataset DS000221[35], which includes semantic relatedness and sound discrimination trials collected from the same participants at 5, 7, and 9 years of age. On each trial, children heard a pair of words and judged whether they were related (Semantic Relatedness) or sounded similar (Sound Discrimination). We extracted trial-level response times and binary accuracy values and used these as behavioral benchmarks.

For each cohort-specific embedding (TO: 18–36 months; 3PLUS: 3–5 years; 5PLUS: 5–12 years; ADULT; and a mixed model assigning 3PLUS similarities to 5-year-olds and 5PLUS similarities to 7- and 9-year-olds), we computed cosine similarity scores for all word pairs and fitted generalized additive mixed models (GAMMs) to accuracy. Each model included smooth terms for age (in months), response time, and cosine similarity, a fixed effect of trial type, and a tensor interaction between age and RT by trial type. We included random intercepts for each participant. A baseline model, without the similarity term, served as a reference. Model fit was compared using Akaike's Information Criterion (AIC), fREML, log-likelihood, and cross-validated log-loss.

The TO embedding achieved the best AIC (17,197), fREML (29,899), and log-likelihood (−8375), followed by 5PLUS, mixed, ADULTS, and 3PLUS. However, in 10-fold cross-validation, the 5PLUS model achieved the lowest log-loss (0.358), with all embedding-informed models out-performing the baseline (0.375).

We also examined the effective degrees of freedom (EDF) of the smooth term on age and the response time–age interaction to assess the complexity of model fits (Table 1). The TO, MIXED, and 5PLUS models yielded low EDF values for the age smooth (TO: 2.21; 5PLUS: 2.40; MIXED: 2.48), indicating that the cohort-specific similarity scores accounted for much of the age-related variance in children's responses (Table 2). In contrast, the 3PLUS and ADULTS models showed higher EDFs (approximately 8), suggesting that, without an experience-related representational proxy, the models required greater flexibility to account for developmental change. As a reminder, higher EDF values indicate that the model estimated a more flexible, nonlinear age effect.

We fitted the same set of generalized additive mixed models to children's performance on the sound discrimination task. Model comparisons replicated the ranking observed in the semantic task. The 5PLUS model again achieved the best overall fit, with the lowest log-loss (0.425), second-highest likelihood (−12,164), second-lowest fREML (38,925), and second-lowest AIC (24,879). The TO model slightly outperformed it on AIC (23,683), fREML (36,970), and log-likelihood (−11,571). The baseline

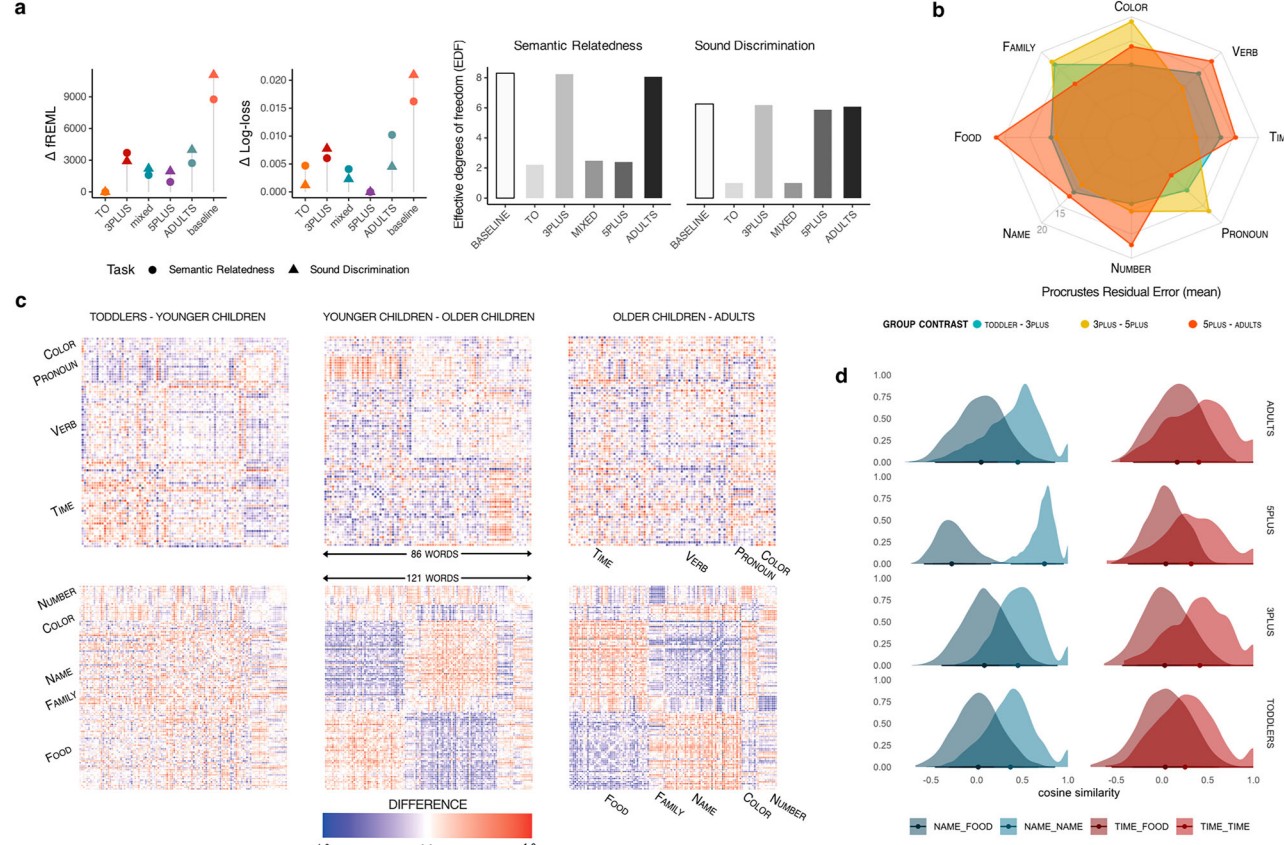

**Fig. 3 | U-shaped development of semantic distance between word categories.** Cohort-based embeddings align with behavioral performance (**a**) and reveal that category structure changes in a non-monotonic pattern over development (**b–d**). **a** Model comparison of cohort-trained CBOW embeddings against children's behavioral responses in the OpenNeuro DS000221 dataset. Lollipop plots show fREML and cross-validated log-loss for semantic relatedness and sound discrimination tasks at ages 5, 7, and 9. All cohort-based embeddings outperformed the baseline. TO (1.5–3 years) model fit best in-sample (AIC, fREML, log-likelihood) and the 5PLUS (5–12 years) model generalized best in cross-validation (log-loss).

TO, MIXED (age-matched mix of 3PLUS and 5PLUS), and 5PLUS models also showed simpler age smooths, indicating that their representations accounted for developmental change more directly than 3PLUS and ADULT models in the semantic task. **b** Mean Procrustes residual error between embeddings across cohorts. Higher values indicate greater differences in representational structure. **c** Difference matrices of Euclidean distances between target categories across age groups. **d** Distributions of cosine distances within and across category boundaries. Distance patterns for word pairs such as name-food (volatile vs. stable) and time-food (abstract-concrete) reflect cohort-specific shifts in representational geometry.

## Table 1 | Effective degrees of freedom and smooth term significance for age

| Model | Semantic relatedness | | | Sound discrimination | | |
|---|---|---|---|---|---|---|
| | EDF | Chi-square | p value | EDF | Chi-square | p value |
| TO | **2.21** | 38.66 | < 0.001 | **1.00** | 44.17 | < 0.001 |
| 3PLUS | 8.25 | 76.29 | < 0.001 | 6.19 | 60.85 | < 0.001 |
| 5PLUS | 2.40 | 40.27 | < 0.001 | 5.87 | 56.62 | < 0.001 |
| ADULTS | 8.07 | 89.99 | < 0.001 | 6.08 | 66.55 | < 0.001 |
| MIXED | 2.48 | 40.43 | < 0.001 | 1.00 | 36.71 | < 0.001 |
| BASELINE | 8.30 | 106.45 | < 0.001 | 6.26 | 71.32 | < 0.001 |

Estimated effective degrees of freedom (EDF), Chi-square statistics, and p values for the smooth term $s(age\_months)$ in the GAMMs predicting performance in the semantic relatedness and sound discrimination tasks. Lower EDF indicates smoother (less flexible) age-related effects. Boldface indicates the lowest EDF values within each task.

model performed worst across all metrics (AIC = 31,643, fREML = 48,052, log-loss = 0.446).

The similar pattern of results observed across the semantic and sound discrimination tasks supports the conclusion that the cohort-specific word2vec models captured key aspects of the representational structure available to children, as these models generalized across distinct types of

judgments. While AIC and fREML indicated slightly better in-sample fits for the toddler (TO) model, a minimal difference in cross-validated log-loss suggested that the 5PLUS models may offer better generalization to the behavioral responses of a specific cohort of children in this task setting. Given that the underlying corpora are heterogeneous and were collected across children and contexts over several decades, these results should be interpreted as approximate but robust indicators of cohort-specific structure rather than as precise estimates of generalization performance.

Across all models, the response time–age interaction was most flexible on unrelated trials, with EDFs ranging from 7.84 to 9.40 in the semantic relatedness task and from 4.23 to 5.33 in the sound discrimination task. In the semantic task, EDFs for unrelated trials were 8.08 (TO), 8.09 (3PLUS), 7.84 (5PLUS), 8.38 (ADULTS), 8.54 (MIXED), and 9.40 (baseline). In the sound discrimination task, EDFs were 4.23 (TO), 5.11 (3PLUS), 5.33 (5PLUS), 5.24 (ADULTS), 5.09 (MIXED), and 5.03 (baseline).

For completeness, the baseline GAMMs showed similar trial-type-specific flexibility. In the semantic discrimination task, EDFs for the interaction were 2.08, 4.70, and 9.40 for high-association, low-association, and unrelated trials, respectively (Chi-square = 13.31, $p$ = 0.0032; 12.18, $p$ = 0.0608; 89.14, $p$ < 0.0001). In the sound discrimination task, EDFs were 1.61, 4.80, and 5.03 for onset, rhyme, and unrelated trials, respectively (Chi-square = 3.00, $p$ = 0.252; 35.71, $p$ < 0.0001; 50.58, $p$ < 0.0001).

The interaction between age, response time, and accuracy showed distinct developmental patterns. Five-year-olds performed better on related

**Table 2 | Model comparison statistics for cohort-trained language models**

| Model | Semantic discrimination | | | | Sound discrimination | | | |
|---|---|---|---|---|---|---|---|---|
| | AIC | fREML | LogLik | Log-loss | AIC | fREML | LogLik | Log-loss |
| TO | **17,197** | **29,899** | −8375 | 0.363 | **23,683** | **36,970** | −11,571 | 0.426 |
| 3PLUS | 19,344 | 33,596 | −9434 | 0.364 | 25,791 | 39,880 | −12,620 | 0.432 |
| 5PLUS | 17,569 | 30,839 | −8561 | **0.358** | 24,879 | 38,925 | −12,164 | **0.425** |
| ADULTS | 18,943 | 32,620 | −9235 | 0.369 | 26,348 | 40,948 | −12,891 | 0.429 |
| Mixed | 18,082 | 31,497 | −8816 | 0.362 | 25,123 | 39,175 | −12,293 | 0.427 |
| Baseline | 22,651 | 38,659 | −11,084 | 0.375 | 31,643 | 48,052 | −15,545 | 0.446 |

Model fits for the generalized additive mixed models (GAMMs) predicting children's behavioral responses in the semantic relatedness and sound discrimination tasks. For each cohort-specific embedding (TO = 18–36 months, 3PLUS = 3–5 years, 5PLUS = 5–12 years, ADULT = adult corpus, MIXED = combined mapping of 3PLUS and 5PLUS), the table lists Akaike's Information Criterion (AIC), restricted maximum likelihood (fREML), log-likelihood, and ten-fold cross-validated log-loss. Lower AIC, fREML, and log-loss indicate better model fit. The best value for each metric is highlighted in bold.

**Table 3 | Linear fits of log-frequency by rank for parts of speech and utterance positions across cohorts**

| Cohort | Feature | n | df | Slope ± SE | t | 95% CI | p | $R^2$ |
|---|---|---|---|---|---|---|---|---|
| TO | Parts of speech | 1 | 12 | −0.210 ± 0.008 | −26.1 | [−0.228, −0.193] | < 0.0001 | 0.983 |
| TO | Utterance position | 1 | 48 | −0.260 ± 0.007 | −35.4 | [−0.275, −0.245] | < 0.0001 | 0.963 |
| 3PLUS | Parts of speech | 1 | 12 | −0.180 ± 0.008 | −23.0 | [−0.197, −0.163] | < 0.0001 | 0.978 |
| 3PLUS | Utterance position | 1 | 48 | −0.238 ± 0.004 | −53.7 | [−0.247, −0.229] | < 0.0001 | 0.984 |
| 5PLUS | Parts of speech | 1 | 12 | −0.184 ± 0.011 | −17.0 | [−0.207, −0.160] | < 0.0001 | 0.960 |
| 5PLUS | Utterance position | 1 | 48 | −0.245 ± 0.003 | −83.3 | [−0.250, −0.239] | < 0.0001 | 0.993 |
| ADLT | Parts of speech | 1 | 12 | −0.211 ± 0.012 | −17.3 | [−0.238, −0.185] | < 0.0001 | 0.961 |
| ADLT | Utterance position | 1 | 48 | −0.209 ± 0.001 | −147.0 | [−0.212, −0.206] | < 0.0001 | 0.998 |

Each row reports the regression slope (estimate ± SE), test statistic, p value (reported as p < 0.0001 for all models), 95% confidence interval, degrees of freedom, sample size n, and model fit ($R^2$). Parts-of-speech frequencies and utterance-position frequencies both show close-to-geometric distributions with highly consistent cohort-specific slopes.

trials when they responded slowly but showed lower accuracy at short latencies, suggesting the benefits of deliberation. In contrast, 9-year-olds showed the opposite pattern: they were more likely to respond accurately when reacting quickly, with performance declining at longer latencies across all trial types. This points to a downside of deliberation in this type of task, and suggests that once representations are established, taking more time may hinder rather than help [cf. [47]]. Moreover, the pattern we observe suggests that judging relatedness and rejecting unrelated pairs may at times engage distinct processes, rather than being the result of the same simple binary decision process.

**Learning redistributes and smooths information in sequences**
To address our third research question, whether deviations in production patterns align with structural model differences, we analyzed the distribution of parts of speech and the distribution of sequence positions. Parts of speech approached a geometric distribution in all cohorts, with excellent linear fits between log-frequency and rank (TO: $R^2 = 0.983$; 3PLUS: $R^2 = 0.978$; 5PLUS: $R^2 = 0.960$; ADLT: $R^2 = 0.961$; see Table 3). The distribution of utterance positions showed a similarly close fit (TO: $R^2 = 0.963$; 3PLUS: $R^2 = 0.984$; 5PLUS: $R^2 = 0.993$; ADLT: $R^2 = 0.998$; see Table 3).

Geometric distributions are memoryless: they maximize the probability that speakers sampling from them will all be able to develop and maintain sufficiently similar models of their probabilities, irrespective of the differences in individuals' experience. They also describe the Shannon channel-capacity limit for discrete sequences, marking the point at which a system conveys maximal information under minimal structural constraints[48].

Differences between older children and adults were reflected in the means and slopes of these distributions. Learning appeared to affect the parameters but not the overall distribution type. Rate parameters represented the change in termination probability at each sequence step and, for parts of speech, the probability of predicting the next word category given no prior information. Although distributions in older children and adults closely approximated the geometric form, the rates at which utterances terminated and syntactic contexts were updated-varied. For sequence-position distributions, slopes flattened with age (TO: −0.260, 3PLUS: −0.238, 5PLUS: −0.245, ADLT: −0.209), and variability decreased (SD = 3.86 in TO, 3.05 in ADLT). In contrast, part-of-speech slopes (TO: −0.210, ADLT: −0.211) and dispersion (SD = 0.89–0.90) remained stable. As rate parameters change, self-information at each sequence step also changes: the entropy of segmentation rates increased from 2.4 bits in toddlers to 3.7 bits in adults, whereas the entropy of parts of speech remained constant ($H = 3.4$) across age groups (Fig. 4a). The syntactic structure of children's utterances seems to develop with the average amount of information children can manage at once [cf. [49]].

While the overall distribution of parts of speech is similar across cohorts, the probabilities of individual categories change (see Supplementary Fig. S7). This reranking of part-of-speech probabilities reflects a redistribution of syntactic categories in sequences, while the global sampling-invariant (geometric) distribution is preserved. Suppose that speakers' models of grammar are the same (i.e., children aim to produce adult-like sequences). In this case, reranking results from subtle adjustments in word order and utterance length, which maintain a stable transmission of information in time [channel capacity, cf. [50]].

Suppose that words in sequences serve to manage the uncertainty about the upcoming words. From this perspective, the uncertainty about the overall message (and possible future words) tends to decrease as the sequence progresses, while the likelihood of sequence termination increases. By implication, the sequence position can gauge the uncertainty associated with the context and act as a minimal representation of context. With a perfect correlation between sequence length and sequence probability, self-information at any sequence position is co-determined by the termination probability. In older children and adults, the entropy associated with

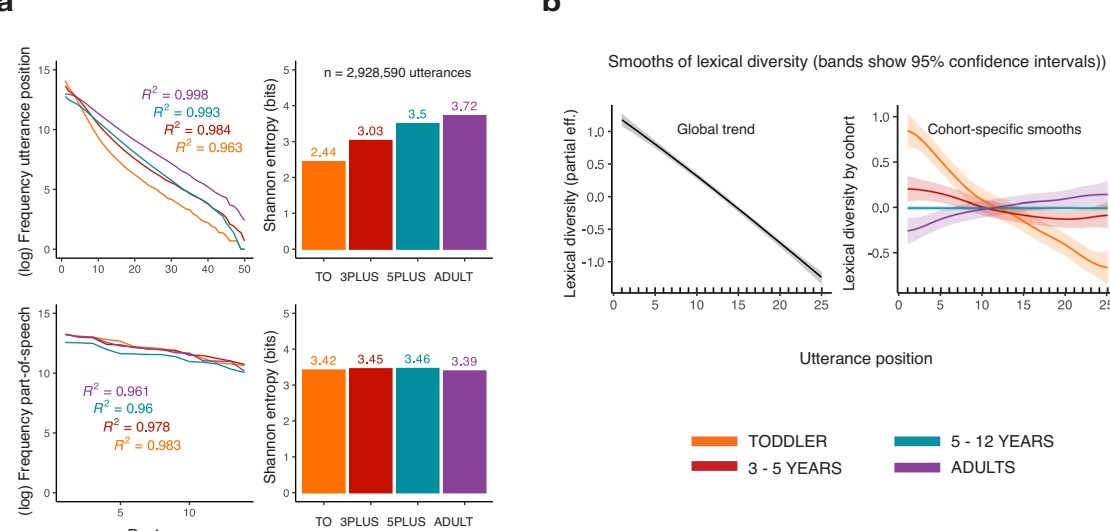

**Fig. 4 | Children produce longer and more variable sequences with age, increasing segmentation entropy and shifting how information is distributed across utterances. a** With speaker age, the fit to a geometric distribution improves, and the slope becomes shallower. This redistribution smooths the distribution of information across sequences, and increases segmentation entropy. Meanwhile, the parameters of the parts-of-speech distribution are maintained. **b** Lexical diversity by position (left) and cohort-specific factor smooths (right). Older cohorts reduce lexical diversity in initial positions and increase diversity in final positions.

**Table 4 | Parametric effects for developmental models of MI, first-order JSD, and second-order JSD**

| Measure | Cohort | $\beta$ | SE | 95% CI | $t$ | $p$ |
|---|---|---|---|---|---|---|
| **Mutual information (MI)** | | | | | | |
| Intercept (TO) | – | 0.2787 | 0.0089 | [0.261, 0.296] | 31.35 | < 0.001 |
| 3–5 years | vs. TO | 0.0796 | 0.0125 | [0.055, 0.104] | 6.36 | < 0.001 |
| 5–12 years | vs. TO | 0.0881 | 0.0125 | [0.064, 0.112] | 7.04 | < 0.001 |
| Adults | vs. TO | 0.1461 | 0.0127 | [0.121, 0.171] | 11.54 | < 0.001 |
| **First-order JSD** | | | | | | |
| Intercept (TO) | – | 0.0721 | 0.0017 | [0.069, 0.075] | 42.85 | < 0.001 |
| 3–5 years | vs. TO | 0.0180 | 0.0024 | [0.014, 0.022] | 7.62 | < 0.001 |
| 5–12 years | vs. TO | 0.0235 | 0.0024 | [0.019, 0.028] | 9.93 | < 0.001 |
| Adults | vs. TO | 0.0351 | 0.0024 | [0.030, 0.040] | 14.65 | < 0.001 |
| **Second-order JSD** | | | | | | |
| Intercept (TO) | – | 0.0752 | 0.0013 | [0.073, 0.078] | 59.70 | < 0.001 |
| 3–5 years | vs. TO | 0.0151 | 0.0018 | [0.012, 0.018] | 8.49 | < 0.001 |
| 5–12 years | vs. TO | 0.0234 | 0.0018 | [0.020, 0.027] | 13.20 | < 0.001 |
| Adults | vs. TO | 0.0474 | 0.0018 | [0.044, 0.051] | 26.35 | < 0.001 |

Toddlers (TO) are the reference level. Reported are coefficient estimates ($\beta$), standard errors (SE), 95% confidence intervals (CI), $t$ statistics, and $p$ values.

sequence segmentation or event rate increased as the average sequence lengthened (Fig. 4a). The self-information and entropy for part-of-speech categories remained constant. These differences between the distributions suggest that speakers learn to manage the increasing entropy by efficiently redistributing words in sequences. That is, the structure of sequences and the early representations of syntax appear to develop in parallel with word learning, the changes in the uncertainty associated with individual categories that follow word learning, and the development of working memory/cognitive control that allows more variability in segmentation.

### Developmental changes are most prominent at sequence boundaries

To examine how this redistribution of uncertainty plays out in children's productions, we analyzed adjacent-category mutual information (MI) and Jensen–Shannon divergence (JSD). MI measures how much knowing the current part-of-speech category reduces uncertainty about the next one (i.e., how strongly categories are tied together in local grammatical dependencies). JSD quantifies how different the distribution of categories at a given position is from the overall baseline: first-order JSD compares the position-specific distribution of individual categories to the baseline, while second-order JSD does the same for category-to-category transitions.

Both measures showed a consistent developmental pattern. For mutual information (MI), toddlers' transitions were the least informative, with increasing values in 3–5-year-olds, 5–12-year-olds, and adults (see Table 4). Jensen-Shannon divergence (JSD) showed a parallel pattern across both first- and second-order distributions, again with toddlers showing the lowest divergence and adults the highest (Table 4). Importantly, both first- and second-order models captured pronounced position effects, with the strongest cohort-specific smooths at utterance boundaries (all smooth terms: $F \geq 2.11$, $p \leq 0.001$). The second-order model explained the most

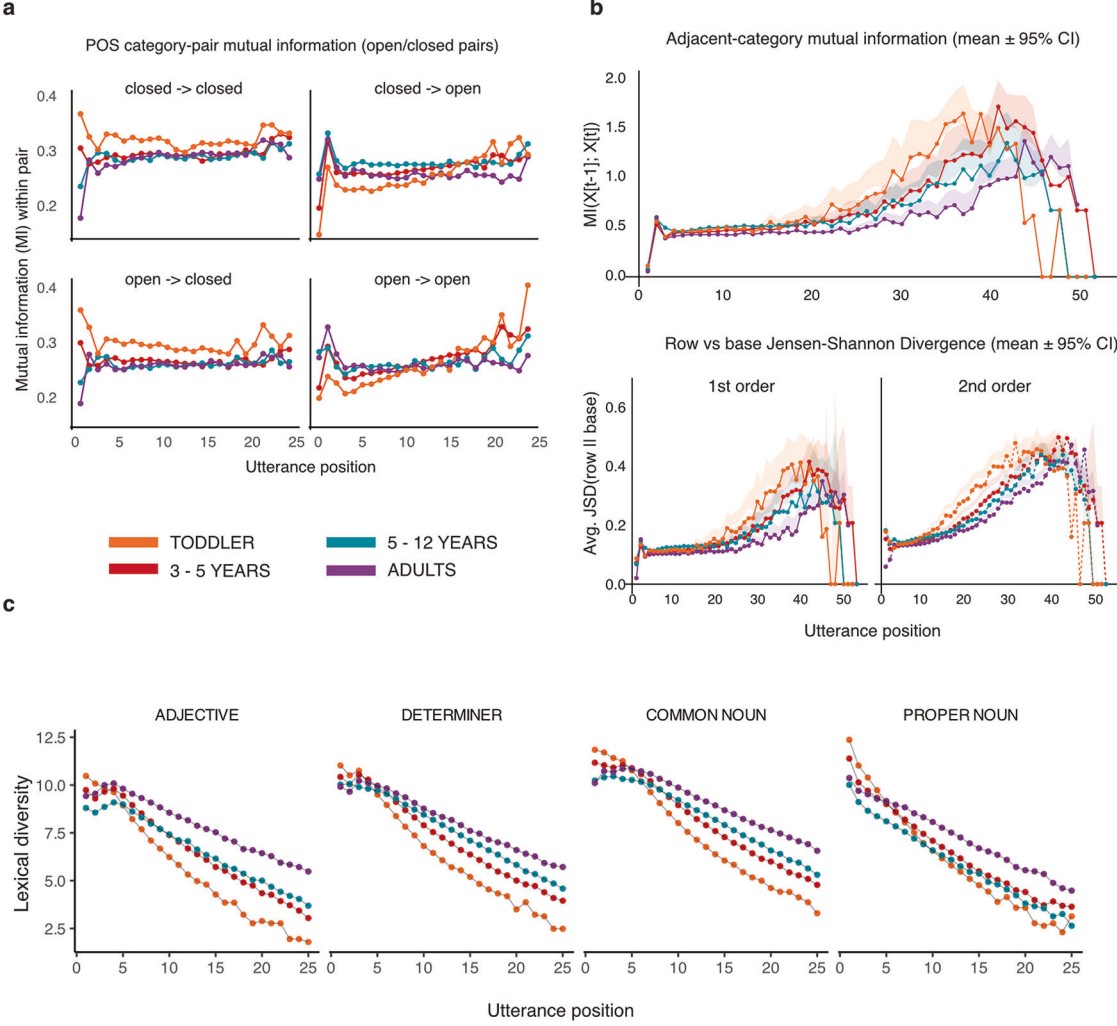

**Fig. 5 | Learning reorganizes sequences and leads to a redistribution of information towards the utterance boundary. a** Adjacent-category mutual information at transitions between open- and closed-class word categories. In younger children, transitions to closed-class words are more informative. At transitions to open-class words, information increases linearly-starting lower than in older speakers and rising higher in later utterance positions. In older speakers, mutual information is more evenly distributed across word classes and utterance positions. **b** Adjacent-category mutual information (top) and Jensen-Shannon Divergence (JSD) from baseline category distributions (bottom), by position and cohort. Cohort differences average out in the aggregate and are only present at utterance boundaries. JSD quantifies the difference between position-specific part-of-speech (POS) distributions (row) and the overall distribution (base). First-order divergence accounts for position only, and second-order divergence for position + previous POS. With age, information becomes more evenly distributed across utterances. **c** Cohort-differences in lexical diversity of determiners, adjectives, and common and proper nouns across utterance positions.

variance (adj. $R^2 = 0.96$), followed by first-order JSD (adj. $R^2 = 0.89$) and MI (adj. $R^2 = 0.83$).

Lexical diversity decreased almost linearly across utterance positions, as indicated by the global smooth term (EDF = 1.73, Ref.df = 2.06, $F = 510.20$, $p < 0.001$), showing a strong downward trend with only minimal curvature. Cohort-specific factor-smooths showed systematic adjustments around this shared trend. Toddlers exhibited a steeper early decrease (EDF = 6.18, $F = 23.65$, $p < 0.001$), 3–5-year-olds showed a smaller deviation from the global smooth (EDF = 3.21, $F = 1.95$, $p < 0.001$), and adults displayed intermediate flexibility (EDF = 3.37, $F = 1.94$, $p < 0.001$). The 5PLUS smooth did not differ significantly from a flat adjustment (EDF ≈ 0, $p = 0.36$), indicating that this cohort followed the global decline most closely.

Consistent with the modeled smooths, lexical diversity was highest for toddlers at the utterance-initial position, followed by 3–5-year-olds and 5–12-year-olds, whereas adults showed the lowest diversity at onset. This pattern reversed at later positions, where adults and older children produced the most context-specific and varied words (Fig. 4b). The effect was more pronounced in productive word categories than in closed-class categories (Fig. 5c and Supplementary Table S4).

## Discussion

We presented a systemic analysis of structural reorganization in utterances produced by children of varying ages and examined the effects of this reorganization on semantic embeddings trained on these utterances. Our findings reveal that children's production patterns reflect progressively detailed representations of language structure. Initially, children start with dense, less differentiated representations, but as they learn more about the distinctions between word categories, these representations evolve into more complex and sparse systems of relations. The increase in complexity and sparseness leads to a U-shaped dynamic in the development of the model's category structure, a pattern consistent with empirical observations of development.

We understand the U-shaped pattern in the model as follows: the model extracts features based on word co-occurrence patterns. In our corpus, morpho-syntactic and prosodic patterns are implicitly folded into surface word co-occurrence statistics (i.e., different dimensions of the speech signal do not develop independently from each other). When the speech register shifts from child to adult discourse, interactions among these dimensions become more complex. A low-dimensional embedding

reorganizes its geometry, producing an apparent baseline shift and a U-shaped trajectory in model estimates. We suggest that this reflects the way that children's representations change systematically in response to the properties of the multi-level distributional structures from which they learn.

Critically, within the subcategories of words, the patterns of development in the models match the observed differences in how children learn and master word usage. These developmental trajectories suggest that learning to differentiate between categories of words is closely tied to changes in the temporal structure of sequences—and that learning the relational structure of communication systems affects the rate and order in which phrases, words, and sounds naturally unfold in spontaneous speech. As children's language skills mature, their command over temporal/sequential cues leads to more complex patterns of production distributed across contexts. Our analyses shed light on how these developmental changes contribute to the organization and complexity of children's language models over time.

Previous research using vector space analyses has shown that child-directed speech is structured in ways that support inference, even in the absence of fully developed adult-like syntactic knowledge[30]. Our results suggest that learning from sparse distributions facilitates this flexible, dynamic framing. In this process, a learner's uncertainty at any developmental stage is tied to shifts in base rates that distinguish between context (predictive structures) and content (distributional inferences drawn from these structures). We suggest that adult knowledge of category structures enables experienced speakers to adjust this base rate dynamically, distinguishing between aligned patterns (with entirely predictable distributions) and irregular or embedded dimensions that are sparsely distributed across space and time.

The use of color and number words in child and adult speech exemplifies how base rate shifts affect patterns of generalization (category learning) and word learning. In children's speech, color words are primarily used literally, to describe colors[51]. With learning, the use of color labels and other modifiers becomes increasingly conventionalized; their function tilts towards enhancing the predictability of specific sets of associated nouns and redistributing the information within sequences across contexts[52,53]. This move to using color words in more grammatical ways illustrates how later learning builds on initially learned patterns and sequential relations and how word order increasingly becomes important to grammatical structure. Stable predictive relations between words are a key aspect of communication, forming the foundation of each users' grammar. Learning this grammar leads to differences in the rates at which the entropy of word segments in fixed grammatical frames (e.g., determiners and nouns in noun phrases) increases as speakers learn new words. In contrast, number words appear to follow a qualitatively different pattern, suggesting a distinct path of grammaticalization and learning.

We used cohort differences in lexical diversity and conditional density in sequences to gauge the protracted convergence of learning words from different lexical categories. Our results support the idea that once words from closed categories and common syntactic frames are learned, speakers adjust their utterances to account for the growing number of words they know and the changes in the probabilities that words will follow one another. Moreover, once syntactic patterns are learned, they create stable and predictable frames that allow for greater variability within those frames. The increased flexibility suggests that, as children's understanding of syntactic structures improves, they can explore and produce a broader range of lexical items within these stable frameworks. Simultaneously, learning new items destabilizes/increases the uncertainty over existing frames as the variability in transitional probabilities increases. This ought to lead to more variability in articulation and word order. In this sense, the learning of syntax and semantics is a gradual extraction of increasingly variable, context-specific structures from latent regularities in speech sequences.

As speakers' grasp of category structures improves, their patterns of word use adapt accordingly. This development is particularly evident in the contexts surrounding number words. When children use number words, they frequently embed them within number lines or counting sequences, such as *the spider has two three four five legs* or *I'm gonna have one two three four five six* [examples from CHILDES[54]], such that number lines are among the most common n-grams in the English portion of the CHILDES database. Additionally, when numbers are used as quantifiers, like "the giraffe has four legs," they are often verbatim repetitions of preceding adult utterances.

By contrast, adults often use number words in more complex scenarios, such as discussing time (*we had lunch two hours ago*) or specific quantities (*You will need six medium fresh peaches for a peach cobbler*). This difference suggests that number lines may serve as an early form of sequence grammar for children, providing a foundation that gradually develops into more complex linguistic structures as their language models become more structured and sparse [cf. 5,7,55, for different perspectives on the processes involved]. To better explain this, consider the relationship between six and seven. Six is a strong cue for seven in number lines, as seven frequently follows six. However, seven does not predict six. Temporal order regularities create asymmetrical predictive relationships between word pairs. As children encounter phrases like "six minutes," "six days," or "six o'clock," they start to unlearn the relationship between six and seven in favor of the competing outcomes. Simultaneously, the relationship between seven and six becomes less skewed, as both serve as contextual cues to time words and countable items. Over time, six and seven become more reliable cues to a more variable class of nouns, and through this, they become more similar. Additionally, as our results suggest, the size of the context window adjusts, helping to distinguish fixed sequences, such as "five, six, seven, eight", from more flexible frames such as "I'll be back in six [minutes, days, months]." Flexible switching between these two qualitatively different tasks, counting and reasoning about magnitudes, is an important feature of human cognition and seems to deploy distinct neural mechanisms[56].

When framed in terms of information[48], the distribution of cognitive and neural resources can be seen as responding to the distribution of contrastive features across multiple discriminative domains organized by a shared structure [cf. 57, on the notion of alignable differences]. In languages, such distributions preserve contextually relevant differences [cf. 58,59]. As speakers become more skilled at distinguishing between various types of communicative signs, including knowing when not to use them, the informational affordances shift towards distinguishing between domain-specific dimensions (e.g., discriminative features of particular subsets of words rather than any words). This interpretation is consistent with the developmental patterns we observe in children's productions.

In the absence of certainty or a conceptual map, utterance structure provides a temporal framework for reallocating attention: information may be spread over longer spans when uncertainty is high, and concentrated locally when the context allows. Because tasks differ in scope and affordance, they can be executed in different ways depending on the speaker's current model, drawing on different resources; superficially similar behaviors need not rely on the same underlying processes [cf. 60]. Moreover, verbal instruction can shift children's strategies [as shown in the Dimensional Change Card Sort[61]]. Together, these observations suggest that the structure of linguistic sequences may serve as a proxy for, and at times a guide to, the developing ability to redirect attention.

Against this backdrop, our analyses do not support simple developmental claims such as 'nouns are learned before verbs.' The patterns we observe are better described as outcomes of a dynamic, multilevel process that is co-determined by grammatical constraints on ordering, the temporal organization of utterances (including variable-length frames and boundary effects), and the skewed distribution of words in the input. That environmental skew itself can change over time and across registers, and categories can differ in how their subcategories reorganize in response: some relying more on context-sensitive frames; others relying more on lexical contrast; and others depending on both. In addition, the extent to which communicative codes align (or misalign) with a child's or adult speaker's model will vary by context and task, introducing volatility that can increase or decrease apparent category differences in a multiplicative fashion over time. Accordingly, our observations emphasize the importance of information

and how it spreads or becomes concentrated over time, as opposed to supporting the idea of fixed trajectories for specific parts of speech. Rather, these findings expose patterns that point toward shifts in the dynamics of attention and cognitive strategies that are related to the dynamic reallocation of attention[62,63]. Surface statistics alone cannot settle how this shift unfolds.

These preliminary conclusions about how grammatical systems develop align with observations that children's language use progresses from simple, often repeated sequences to more flexible and complex patterns[64–66]. Our results suggest that an important factor in this transition is the cumulative effect of learning in sparse environments, which shapes the distribution of error signals that support discriminative learning and generalization across sequences with invariant structures. As children's speech develops, the signal becomes more detailed, with new information emerging at the intersection of uncertainty and context, specifically, at points where errors are informative because contextual uncertainty is not at its extremes [cf. [67]]. Simultaneously, more predictable, structural dimensions of speech sequences become less variable. We suggest that protracted learning from input that is unevenly distributed across various contexts affects the distribution of error and information in speech sequences.

### Limitations
We noted above that children's use of context in word learning has, to date, been little studied, with researchers often focusing on how children learn word meanings in isolation. By contrast, apart from allowing us to sidestep the thorny question of whether children ever learn word meanings in the way that researchers assume[68], this study of the way children master systems of words raises important questions about how learning these systems interacts with other dimensions of communication, including acoustics, timing, gesture, and gaze, which play an important role in early learning.

Children are highly responsive to prosody in sequences, with prosodic variations playing a crucial role in interpreting input during early language learning[69–71]. As children's understanding of syntactic and semantic categories begins to align with adult-like patterns, how does their response to variability and error in the input evolve? Specifically, do responses to variable errors in the stable dimensions, such as pauses and vowel duration, differentiate as children learn and categorize language structures? Children's sensitivity to prosody changes with experience[72–74]. Once children have learned the category structure, a richer selection of context-specific acoustic features becomes accessible. These features include subtle changes in articulation patterns and greater variation in temporal resolution and base rates, carving out more nuanced and densely populated dimensions of representation space. In adult speech, production patterns reveal sensitivity to the distribution of information in linguistic systems; variability in acoustic features becomes concentrated at the utterance boundaries as learning progresses, marked by hyperarticulation of more informative sounds and hypoarticulation of less informative sounds[58,75,76]. Do the development of acoustic contrasts in infants' and children's early speech have similar properties? Do early articulations and responses to prosodic variation reflect the organization of children's language models?

Our findings show that cohort differences reflect how speakers distribute information across variable-length spans of speech, with production patterns shaped by each learner's evolving model of adult sequences. Learning emerges from the misalignment between understanding and execution, creating shifts in informational affordances that co-determine which dimensions are treated as informative. This reframes developmental change as an interaction between environmental affordances and the learner's current model, influencing cognitive strategies and brain network activity. Because speech transcripts capture only linguistic codes, converging evidence from eye-tracking and brain measures is needed to test how linguistic uncertainty shapes attention through the temporal alignment of speech with sensory processing of objects and events. These pressures can refine models of communication in some contexts and blur them in others, while carving out or obscuring models of the world as words and the world compete for informativeness.

Another important limitation of this study is the scarcity of large age-labeled datasets that could allow more rigorous testing of our hypotheses. The limited availability of such data sets restricts our ability to validate our findings across a wider range of developmental stages. In this study, we tested our models on held-out sets to ensure robustness. Still, the results highlight a critical gap: We observe substantial differences between the language production of 24-month-olds and 36-month-olds. This period is marked by rapid language development, and much learning and structural reorganization occur within this time frame. However, accurately tracking and analyzing the nuances of this developmental phase requires a finer granularity of age clusters and much larger datasets than those currently available.

Accordingly, we suggest that our study be seen as a small step in what will inevitably be a much longer journey.

## Data availability
All data have been made publicly available at OSF (https://osf.io/qxb9j/).

## Code availability
All data analysis scripts have been made publicly available at OSF (https://osf.io/qxb9j/).

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

## Acknowledgements
M.L. was funded by the Max Planck Society. M.R.'s contribution was supported by a grant from the Deutsche Forschungsgemeinschaft (DFG 547529231). The funders had no role in study design, data collection and analysis, decision to publish, or preparation of the manuscript.

## Author contributions
M.L. and M.R. designed research; M.L. performed research; M.L. analyzed data; and M.L. and M.R. wrote the paper.

## Funding

## Competing interests
The authors declare no competing interests.
