## [Transparent Peer Review file · Communications Psychology]

Sequence structure in children's speech reveals non-linear development of relations between word categories

Corresponding Author: Dr Maja Linke

Version 0:

Decision Letter:

Dear Dr Linke,

Thank you for your patience during the peer-review process. Your manuscript titled "Sequence structure in children's speech reveals non-linear development of relations between syntactic and semantic categories." has now been seen by 3 reviewers, whose comments are appended below. You will see that they find your work of some potential interest. However, they have raised quite substantial concerns that must be addressed. In light of these comments, we cannot accept the manuscript for publication, but would be interested in considering a revised version that fully addresses these serious concerns.

We hope you will find the Reviewers' comments useful as you decide how to proceed. Should additional work allow you to address these criticisms, we would be happy to look at a substantially revised manuscript. If you choose to take up this option, please highlight all changes in the manuscript text file, and provide a detailed point-by-point reply to the reviewers.

Editorially, we ask that you provide compelling revisions to justify your conclusions (including through additional analyses, see Reviewer #3) and offer a more in-depth discussion of their theoretical implications (Reviewer #1).

In addition, as the work uses human data, we ask that you declare what ethics framework was in place to guide the research. Please indicate whether you complied with all relevant ethical regulations and include a statement affirming this in the manuscript; ensure the manuscript states the name(s) of the board and/or institution that either approved the study protocol - OR- provided guidelines for study procedures (if protocol approval is not required).

I am attaching a checklist that details critical reporting requirements for the revised manuscript. Please attend to each item and ensure your manuscript is fully compliant. We are requesting that your manuscript aligns with these requirements as this facilitates the evaluation of your manuscript, reducing delays in re-review and potential future acceptance. If your revised manuscript is not aligned with these requests on major issues, such as those concerning statistics, it may be returned to you for further revisions without re-review. Additional information can be found in our style and formatting guide Communications Psychology formatting guide.

If the revision process takes significantly longer than five months, we will be happy to reconsider your paper at a later date, provided it still presents a significant contribution to the literature at that stage.

Please use the following link to submit your

- revised manuscript,
- point-by-point response to the referees' comments,
- cover letter (as a separate document),
- the Editorial Policy Checklist (see below),
- the Reporting Summary (see below), and
- the completed Editorial Request Table (attached):

Link Redacted

Thank you for the opportunity to review your work.

Best regards,

Jixing Li

Jixing Li, PhD
Editorial Board Member
Communications Psychology
orcid.org/0000-0002-5210-6224

REVIEWER EXPERTISE:

Reviewer #1 psycholinguistics
Reviewer #2 psycholinguistics, language development
Reviewer #3 psycholinguistics, modelling

REVIEWER REPORTS:

Reviewer #1 (Remarks to the Author):

This paper presents an interesting investigation into how language development influences distributed semantic representations and how these changes are reflected in speech production patterns using word embeddings. However, I have several concerns:

Terminology Clarity: Several key terms are undefined or unclear. For instance, concepts like “base rate shift” and “contextual inferences” are central but insufficiently explained. Phrases such as “this shift reflects changes in the base rates representing system dimensions with aligned distributions” (p.5) are difficult to parse. Similarly, vague terms like “system structure affects timing” need clarification—what system is being referred to?

Age Group Focus: In section 2.1, the emphasis on “younger cohorts” in Questions 1 and 3 is unclear—why not consider all age groups? Please clarify the reasoning behind focusing specifically on younger speakers.

Connection Between Questions and Results: The relationship between the three guiding questions and the conclusions in sections 2.1–2.7 should be more clearly articulated. For example, before section 2.4, a transition such as “To determine whether production deviations in younger cohorts reflect structural differences in learned models...” would help guide the reader. Additionally, Question 2 (on inferring acquisition order from semantic structure) is not clearly addressed in the experiments.

Figure Labeling Issues: There are errors in figure references, such as incorrect naming (e.g., “Figure A2B”).

Inconsistent Category Representation: Figure 2C only shows 6 categories, while Figure 2A includes 8. Please clarify the rationale.

Lack of Quantification of Word Order: The claim that language models are shaped by distributional patterns and word order (p.6) is interesting, but how word order is quantified in the experiments is not clearly explained.

Causality Assumptions: The explanation on page 8 linking a lack of prosodic and morphological information to a U-shaped effect lacks justification. It would be more accurate to attribute the effect to the co-occurrence-based nature of word embeddings, which differs from real-world word learning. The current causal claim is unsubstantiated.

Lack of Concrete Developmental Insights: The paper would benefit from more concrete and interpretable conclusions. For instance, if word learning rates differ by category, can the order of acquisition (e.g., nouns before verbs) be inferred? More specific examples would help make the findings more accessible.

Reviewer #2 (Remarks to the Author):

Review report on the manuscript entitled “Sequence structure in children’s speech reveals non-linear development of relations between syntactic and semantic categories”

Summary: Using a computational method, the current study revisited the classical topic of the U-shape trajectory of child language development. The authors analyzed spoken language data from CHILDES. They categorized the data according to children’s age so as to examine the developmental trajectory. They connected the holistic patterns of the children’s speech with individual words and phrases so as to assess the complexity of contextual information with the development of child language. The study revealed how English-speaking children reorganize semantic networks with the development of their language. The manuscript was well written and easy to follow.

I am a non-expert on the techniques of computational modeling. So I would let the experts comment on the research methodology. My comments are focused on the theoretical backgrounds, and the study’s empirical contributions to the field.

As discussed in the introduction of the manuscript, the research on U-shape developmental trajectory of child language learning has been extensive although they usually focused on limited numbers of linguistic constructs. I would like to see more discussions on the theoretical implications of the current study. I am especially interested in discussions on how the study “identifies the distributional factors that influence the development of early mental models and offer new perspectives on conditions that determine how learning evolves”. As noted by the authors, previous studies on the U-shape trajectory of child language development usually focused on specific linguistic constructs, discussing how children mapped meanings to abstract phonological forms. To contrast, the mental model theory stresses the process of mapping abstract linguistic symbols to specific objects and events. At the end of the second paragraph of the introduction section, the authors discussed that children’s language development mismatched their development of mapping the abstract linguistic symbols with real-world events. One major finding of the study is the increased complexity of the children’s category structure, which was summarized at the beginning of the discussion section. Does this mean that children’s mental models have more and more details with the increase of their age? In addition, the authors found a U-shape developmental trajectory of the children’s holistic language development. Can we say that the development of mapping between language and mental models shows a U-shape trajectory? It will be very interesting if the authors discuss more on how the findings relate to the theoretical frameworks of mental model.

My second comment is about the modeling methods of the study. I feel the authors may need to provide more information on how they selected specific models, if the modeling methods were used in previous research on similar and/or different topics, and how their methods differ from those non-embedding models. These discussions will be very helpful for the readers to understand the methodological contribution of the current study.

Lastly, a minor point: In 4.3, the authors mentioned that they used word2vec models as a baseline, which “differs from other factorization methods like topic modeling or more conventional forms of latent semantic analysis”. The authors may need to cite one or two studies using factorization methods as examples.

Reviewer #3 (Remarks to the Author):

- What are the major claims of the paper? Are they novel and will they be of interest to others in the community and the wider field?

The major claim of the paper is that changes in the distributional geometry of word embeddings, trained on children’s spontaneous speech at different ages, reflect the evolving internal organization of semantic and syntactic categories during development. The authors further argue that these developmental dynamics exhibit a U-shaped pattern and suggest that early utterances are structured by temporary grammars that optimize information distribution. These claims are novel in their methodological execution and conceptual framing, combining distributional semantics with developmental theory in an original way. The paper is likely to be of interest to researchers in developmental psycholinguistics, cognitive science, and computational language modeling.

- Is the work convincing, and if not, what further evidence would be required to strengthen the conclusions?

While the analytic results are technically sound and the developmental patterns observed are consistent, the central interpretation—that embedding space geometry reflects children’s internal cognitive models—requires stronger justification. The authors rely on the assumption that speech co-occurrence patterns directly mirror cognitive representations, but this is not empirically validated in the paper. Further support, such as behavioral data on category confusability, developmental benchmarks, or alignment with known learning trajectories, would be needed to strengthen the causal interpretation. Additionally, showing robustness across different embedding methods or controlling for confounding variables like corpus size and lexical diversity would enhance the argument.

- Do you feel that the paper will influence thinking in the field?

If the assumptions underlying the interpretation of embedding space as a proxy for cognitive structure can be substantiated, this approach has the potential to influence thinking in the field. The work presents a promising framework for linking system-

level changes in language use to representational development, which may inspire further studies using computational tools to analyze developmental corpora. However, the current version of the paper would benefit from more theoretical clarity and empirical support before it can be widely adopted as a model of cognitive development.

- Does the paper represent an advance in understanding which may influence thinking in the field?

Yes, the paper represents a conceptual advance by proposing that language learning can be understood through changes in the structure of distributional semantics over time. It encourages researchers to look beyond isolated word learning and instead consider global structural shifts in the linguistic system. However, the influence of this work will depend on the authors' ability to clearly articulate and justify how their findings map onto cognitive development.

- Does the article present an original study, new analysis, new model, or a direct or extended replication of previous work?

Yes, the study is original in both dataset and methodology. It introduces a new way of modeling and visualizing developmental language change using low-dimensional Word2Vec models trained on age-specific corpora, combined with t-SNE projections, distance metrics, and statistical shape alignment analyses (e.g., Procrustes and Mantel tests).

- Are the data and analysis technically sound? Are they appropriate to answer the research question, e.g., are causal research questions addressed on the basis of causal, rather than correlational evidence?

The analyses are technically competent and methodologically appropriate for a correlational, exploratory study. The authors appropriately use nonparametric and permutation-based methods to examine geometric differences in embedding space. However, the conclusions drawn remain correlational, and the manuscript would benefit from a clearer acknowledgment that no causal inference can be made from the current design.

- Does the paper provide strong evidence for its conclusions?

The evidence is consistent and multifaceted within the scope of the chosen methods, but the conclusions about internal cognitive models are stronger than what the data directly support. The embedding patterns are compelling, but without triangulation from behavioral data or model-independent developmental findings, the cognitive interpretation remains speculative.

- Is the study question important to scientists for a sub-field of psychology?

Yes. Understanding how children organize linguistic knowledge over time is a fundamental question in developmental psychology and psycholinguistics. This study contributes to that discourse by providing a data-driven, systems-level approach to analyzing category structure in child language.

- Are there any special ethical concerns arising from the use of animals or human subjects?

No. The study uses publicly available, de-identified corpora from CHILDES and other sources. There are no direct human subjects or ethical concerns involved.

- Was the study preregistered and if so, did the authors follow the preregistration?

The manuscript does not mention preregistration. While preregistration is not always expected in corpus-based or exploratory modeling studies, some transparency about analytic choices and flexibility would be appreciated.

EDITORIAL POLICIES

We ask that you ensure your manuscript complies with our editorial policies and reporting requirements.

To that end, we require revised manuscripts to be accompanied by two completed items: a reporting summary that collects information on study design and procedure, and an editorial policy checklist that verifies compliance with all required editorial policies

- <https://www.nature.com/documents/nr-reporting-summary.zip>>Nature Research Reporting Summary
- <https://www.nature.com/documents/nr-editorial-policy-checklist.pdf>>Editorial Policy Checklist

All points on the policy checklist must be addressed. Your revised manuscript can only be sent back to the referees if these checklists are completed and uploaded with the revision.

Notes: If you have submitted a Stage 1 Registered Report, Review, Primer, Comment, or Perspective you do not need to submit these forms. If you have already submitted these forms, you may disregard this request.

Version 1:

Decision Letter:

Dear Dr Linke,

Your manuscript titled "Sequence structure in children's speech reveals non-linear development of relations between word categories" has now been seen by our reviewers, whose comments appear below. In light of their advice I am delighted to say that we are happy, in principle, to publish a suitably revised version in Communications Psychology.

We therefore invite you to revise your paper one last time to address the remaining editorial concerns. At the same time we ask that you edit your manuscript to comply with our format requirements and to maximise the accessibility and therefore the impact of your work.

EDITORIAL REQUESTS:

SUBMISSION INFORMATION:

OPEN ACCESS:

* TRANSPARENT PEER REVIEW: Communications Psychology uses a transparent peer review system. On author request, confidential information and data can be removed from the published reviewer reports and rebuttal letters prior to publication. If you are concerned about the release of confidential data, please let us know specifically what information you would like to have removed. Please note that we cannot incorporate redactions for any other reasons.

Link Redacted

Best regards,

Jixing Li

Jixing Li, PhD
Editorial Board Member
Communications Psychology
orcid.org/0000-0002-5210-6224

Marika Schiffer, PhD
Chief Editor
Communications Psychology

REVIEWERS' COMMENTS:

Reviewer #2 (Remarks to the Author):

I have thoroughly reviewed the rebuttal letter and the revised manuscript. I feel that the authors have addressed my comments on the initial draft very well. I appreciate their hard work, and I am happy to approve this revised version.

Reviewer #3 (Remarks to the Author):

Dear Authors and Editors,

I have reviewed the revised manuscript and am satisfied that my concerns have been adequately addressed. I recommend acceptance.

The addition of behavioral validation data in Section 2.4 substantially strengthens the empirical support for your approach. The clarified theoretical framing—positioning embeddings as measurement tools rather than cognitive models—appropriately scopes the claims. I also appreciate the enhanced methodological transparency and explicit acknowledgment of limitations regarding causal inference.

While cross-algorithm validation would have been ideal, I accept your rationale for maintaining a consistent learning framework across age cohorts. The sensitivity analyses and OSF materials provide reasonable evidence of robustness. This work makes a solid contribution to understanding developmental changes in distributional structure and provides a valuable methodological framework for future research.

Response to Reviewer Comments

Reviewer #1 (Remarks to the Author):

This paper presents an interesting investigation into how language development influences distributed semantic representations and how these changes are reflected in speech production patterns using word embeddings. However, I have several concerns:

Terminology Clarity: Several key terms are undefined or unclear. For instance, concepts like “base rate shift” and “contextual inferences” are central but insufficiently explained. Phrases such as “this shift reflects changes in the base rates representing system dimensions with aligned distributions” (p.5) are difficult to parse. Similarly, vague terms like “system structure affects timing” need clarification—what system is being referred to?

Response:

Thank you. We agree this was unclear. We revised the manuscript to remove vague or opaque phrasing; all changes are shown in the tracked-changes manuscript.

Age Group Focus: In section 2.1, the emphasis on “younger cohorts” in Questions 1 and 3 is unclear—why not consider all age groups? Please clarify the reasoning behind focusing specifically on younger speakers.

Response:

We agree the wording was misleading. We do not focus exclusively on younger cohorts. To avoid this impression we (i) added “(as opposed to older speakers)” to Question 1 to make the intended comparison explicit, and (ii) removed the word *younger* from Question 3 and rephrased it to: “Are deviations in cohorts’ production patterns consistent with model differences?”.

Connection Between Questions and Results: The relationship between the three guiding questions and the conclusions in sections 2.1–2.7 should be more clearly articulated. For example, before section 2.4, a transition such as “To determine whether production deviations in younger cohorts reflect structural differences in learned models...” would help guide the reader. Additionally, Question 2 (on inferring acquisition order from semantic structure) is not clearly addressed in the experiments.

Response:

Thank you for raising this point. We have extended the Results to make explicit which analyses address each guiding question.

Question 2 (qualitative changes) is addressed by analyses documenting a reversal of noun–noun relations across ages (Fig. 3C) and by Procrustes-alignment residuals that quantify changes in representational geometry between cohorts. Further analyses in Section 2.4 report changes in adjacent-category mutual information and

Jensen–Shannon divergence from baseline by absolute utterance position (new Fig. 4B–E). Together, these results show a redistribution of information within sequences and a decrease in structural variability, alongside a differentiation of noun categories; concrete examples (e.g., number vs. color words; adjectives vs. nouns) are given on pp. 17–18.

Figure Labeling Issues: There are errors in figure references, such as incorrect naming (e.g., “Figure A2B”).

Response:

Thank you for pointing this out. We corrected all figure reference errors, verified consistency across the main text, captions, and Supplementary Information, and checked and revised every LaTeX cross-reference and link.

Inconsistent Category Representation: Figure 2C only shows 6 categories, while Figure 2A includes 8. Please clarify the rationale.

Response:

Fig. 2C displays six categories chosen because they are distinct from one another and together illustrate different patterns we find when comparing cohort slices. We added a sentence to the Fig. 2C caption to make this explicit and included a larger supplementary figure (Supplementary Fig. A13) showing all eight categories for completeness. The main text now points readers to the supplement.

“Differences between cohorts’ embeddings for six target categories visualized with t-SNE. Distances approximate distances between word vectors; cluster density and distance from the centroid index cohort divergence; the full set of eight categories is shown in Supplementary Fig. A13.”

Lack of Quantification of Word Order: The claim that language models are shaped by distributional patterns and word order (p.6) is interesting, but how word order is quantified in the experiments is not clearly explained.

Response:

Thank you for highlighting this. We have clarified the claim and now explicitly quantify word order in the revised manuscript. Our central point, already summarized in the visual abstract, is that because different types of words serve different functions in sequences, and because transitional uncertainty changes with experience, the developmental trajectory of lexical categories is necessarily uneven.

To make this concrete, we added the following analyses:

- Permutation entropy to quantify sequential unpredictability (Supplementary Analysis A.2);
- Mutual information by absolute sequence position to estimate how informative individual positions are for predicting word categories (Section 2.5, Fig. 4);
- First- and second-order Jensen–Shannon divergence to track how category distributions shift and disperse across age cohorts (Section 2.5, Fig. 4).

These additions are described in the Methods (Section 4.3) and reported in the Results (Sections 2.5 and 2.6, see also Fig. 4 and Supplementary Analyses A.2).

Causality Assumptions: The explanation on page 8 linking a lack of prosodic and morphological information to a U-shaped effect lacks justification. It would be more accurate to attribute the effect to the co-occurrence-based nature of word embeddings, which differs from real-world word learning. The current causal claim is unsubstantiated.

Response:

We agree that our previous wording could be read as claiming that prosody and morphology are external “add-on” cues whose absence directly causes the U-shaped trajectory. Our actual position is systemic rather than modular. Decades of corpus and acquisition work show that morpho-syntactic paradigms, prosodic regularities, and lexical co-occurrence statistics jointly populate the same distributional space; they are informative dimensions of one communicative system, not independent channels (see e.g., Christiansen and Chater, 2016; Wedel, Nelson and Sharp, 2018; Blevins, Ackerman and Malouf, 2016). Because the word2vec model is trained solely on speech transcripts, all morphological and prosodic structure is collapsed into surface word tokens. As the sequences develop, cross-dimensional interactions proliferate; a low-dimensional embedding therefore returns to the dense configuration. We have revised the text to (a) frame this as a plausible systemic explanation rather than a causal claim, and (b) clarify that prosody/morphology are already part of the co-occurrence signal.

We revised the text in section 2.2, which now reads:

*In our corpus, morpho-syntactic paradigms and prosodic patterns are implicitly folded into surface word co-occurrence statistics. When the speech register shifts from child-directed to adult discourse, interactions among these dimensions become markedly richer. A low-dimensional embedding therefore re-organizes its geometry, producing an apparent baseline shift and a characteristic U-shaped trajectory in model performance. We view this as *one systemic explanation* grounded in multi-level distributional structure rather than as a direct causal effect of missing prosodic or morphological cues.*

Lack of Concrete Developmental Insights: The paper would benefit from more concrete and interpretable conclusions. For instance, if word learning rates differ by category, can the order of acquisition (e.g., nouns before verbs) be inferred? More specific examples would help make the findings more accessible.

Response:

Thank you, we appreciate the call for more clarity. We agree that the conclusions needed to be made more accessible, and have revised both the Results and Discussion sections accordingly. Our findings do not address learning rates directly, but rather how category structures reorganize in response to changing distributional

regularities. We suggest that these reorganizations shape (and are shaped by) the certainty with which learners can discriminate categories, reflecting interactions between informational affordances, grammatical constraints, and cognitive development.

We now emphasize that our results suggest discrimination between categories precedes discrimination within them, and that such dynamics may not align fully with absolute surface-level acquisition order claims (e.g., “nouns before verbs”). We also added concrete developmental interpretations: for example, we discuss how natural sequence structure may guide attention, and why some parts of speech show more delayed reorganization. Finally, we propose experimental directions that could test these hypotheses using neurocognitive measures such as brain oscillations and microsaccade rates, and outline how this approach could be applied cross-linguistically.

Reviewer #2 (Remarks to the Author):

Review report on the manuscript entitled “Sequence structure in children’s speech reveals non-linear development of relations between syntactic and semantic categories”

Summary: Using a computational method, the current study revisited the classical topic of the U-shape trajectory of child language development. The authors analyzed spoken language data from CHILDES. They categorized the data according to children’s age so as to examine the developmental trajectory. They connected the holistic patterns of the children’s speech with individual words and phrases so as to assess the complexity of contextual information with the development of child language. The study revealed how English-speaking children reorganize semantic networks with the development of their language. The manuscript was well written and easy to follow.

I am a non-expert on the techniques of computational modeling. So I would let the experts comment on the research methodology. My comments are focused on the theoretical backgrounds, and the study’s empirical contributions to the field.

As discussed in the introduction of the manuscript, the research on U-shape developmental trajectory of child language learning has been extensive although they usually focused on limited numbers of linguistic constructs. I would like to see more discussions on the theoretical implications of the current study. I am especially interested in discussions on how the study “identifies the distributional factors that influence the development of early mental models and offer new perspectives on conditions that determine how learning evolves”.

Response:

Thank you for highlighting this. We have expanded the Discussion section to directly address how our findings help identify the distributional factors that shape early mental models. Specifically, we show that the organization of speech sequences

(how information is distributed across time and across lexical categories) constrains which distinctions can be made, and when.

We suggest that developmental change reflects the evolving alignment between the speaker's internal model and the structured, temporally unfolding input. These dynamics point to a broader theoretical perspective in which early learning is guided not by static representations, but by the informational affordances of dynamic environments. This framework reframes development as a reorganization of attention in response to distributional change, not a linear sequence of learning stages. As such, it offers a generalizable account of early mental model formation—across individuals, languages, modalities, and critically, across developmental time.

As noted by the authors, previous studies on the U-shape trajectory of child language development usually focused on specific linguistic constructs, discussing how children mapped meanings to abstract phonological forms. To contrast, the mental model theory stresses the process of mapping abstract linguistic symbols to specific objects and events. At the end of the second paragraph of the introduction section, the authors discussed that children's language development mismatched their development of mapping the abstract linguistic symbols with real-world events. One major finding of the study is the increased complexity of the children's category structure, which was summarized at the beginning of the discussion section. Does this mean that children's mental models have more and more details with the increase of their age?

Response:

Thank you for this thoughtful question. We agree that the increased complexity of children's category structure over time may appear to reflect more detailed mental models. However, our findings suggest that what develops is not detail in an abstract representational sense, but a more sparse/differentiated model of contextual appropriateness—knowledge about when particular words are used, and when they are not. Children's mental models shift toward more context-specific representations: they learn to omit detail where it is uninformative or inappropriate, and to increase it where context supports finer distinctions (i.e., where uncertainty is not at the extremes). We have clarified this point in the Discussion and frame these developments in terms of the increasing availability of alignable contrasts across communicative contexts, as described in related research on structural alignment. We also added new sections to explicitly address how the link between uncertainty, attention, and representational detail can be studied in relation to both the structure of the learning environment (e.g., sparsity, skewness) and the development of cognitive resources.

In addition, the authors found a U-shape developmental trajectory of the children's holistic language development. Can we say that the development of mapping between language and mental models shows a U-shape trajectory? It will be very interesting if the authors discuss more on how the findings relate to the theoretical frameworks of mental model.

Response:

We appreciate this question and have expanded the Introduction and Discussion to make the links to prior work on mental-model frameworks explicit. While we report U-shaped trajectories in speech behavior, we do not claim that the mapping between language and mental models develops according to a fixed U-shaped pattern. Rather, we interpret the observed U-shape as a surface-level expression of deeper dynamics: a temporary misalignment between the speaker's internal model and the structured input (at this level of description—word co-occurrence), followed by reorganization.

We do not take a definite position on the nature or format of mental models. Instead, we highlight that structured speech sequences (and their measurable reorganization over development) offer a promising way to approach this question empirically. By linking surface behavior to changes in information distribution over time, our approach emphasizes that the expression of mental models may depend on the interplay between neural constraints, cognitive strategies, and the distributional properties of the environment.

Finally, we note that the increased complexity observed in children's productions may reflect greater individual variability in how such models are deployed, rather than uniform increases in representational detail. This may not always be visible in word choice, but it can be inferred from changes in how speakers manage context, ambiguity, and communicative constraints across development. We hope these additions address the request and make the connection clearer.

My second comment is about the modeling methods of the study. I feel the authors may need to provide more information on how they selected specific models, if the modeling methods were used in previous research on similar and/or different topics, and how their methods differ from those non-embedding models. These discussions will be very helpful for the readers to understand the methodological contribution of the current study.

Response:

Thank you for this helpful and constructive comment. We have revised the *Methods*, *Introduction* and *Discussion* sections to clarify the rationale behind our modeling choices and how they relate to prior work.

In brief, we selected a low-dimensional Word2Vec model trained on children's productions, not input corpora, as a way to approximate the child's representational geometry. This builds on earlier work using embeddings to capture lexical category structure and neighborhood dynamics, but differs in three key ways: (i) we use production data as a behavioral signal rather than a proxy for exposure, (ii) we analyze how sequence structure changes over time by slicing the training data by age, and (iii) we intentionally limit dimensionality to expose coarse-grained organizational shifts rather than optimize predictive accuracy.

Lastly, a minor point: In 4.3, the authors mentioned that they used word2vec models as a baseline, which "differs from other factorization methods like topic modeling or

more conventional forms of latent semantic analysis". The authors may need to cite one or two studies using factorization methods as examples.

Response:

We added citations for topic modeling and latent semantic analysis. The sentence now reads:

"This approach differs from factorization methods such as topic modeling (Bergey, Marshall, DeDeo, & Yurovsky, 2022; Roy, Frank, & Roy, 2012) or latent semantic analysis (Alhama et al., 2023; Cassani et al., 2021)."

Reviewer #3 (Remarks to the Author):

- What are the major claims of the paper? Are they novel and will they be of interest to others in the community and the wider field?

The major claim of the paper is that changes in the distributional geometry of word embeddings, trained on children's spontaneous speech at different ages, reflect the evolving internal organization of semantic and syntactic categories during development. The authors further argue that these developmental dynamics exhibit a U-shaped pattern and suggest that early utterances are structured by temporary grammars that optimize information distribution. These claims are novel in their methodological execution and conceptual framing, combining distributional semantics with developmental theory in an original way. The paper is likely to be of interest to researchers in developmental psycholinguistics, cognitive science, and computational language modeling.

- Is the work convincing, and if not, what further evidence would be required to strengthen the conclusions?

While the analytic results are technically sound and the developmental patterns observed are consistent, the central interpretation—that embedding space geometry reflects children's internal cognitive models—requires stronger justification.

The authors rely on the assumption that speech co-occurrence patterns directly mirror cognitive representations, but this is not empirically validated in the paper.

Response:

Thank you for raising this concern. We have revised the Introduction and Methods to make explicit that we use embeddings as a measurement tool to summarize how the distributional structure of productions changes with age.

We adopt a distributed, usage-based view in which conceptual structure emerges from the sequential regularities of language and the temporal constraints on segmentation; individual speakers approximate and exploit this evolving code rather than storing a fixed canonical set of concepts internally. At the same time, conceptual structure is co-determined within a system of interdependent regularities: linguistic codes are not independent of other signals, and children learn

correspondences between words and the world. Because sequential structure is comparatively well defined in our data, we use it as a practical interface to this broader system when quantifying developmental change.

Methodologically, we train the same low-dimensional word2vec network on successive age-restricted corpus slices while holding all parameters constant. Accordingly, differences between spaces arise from age-related changes in word-sequence statistics. It is the structure of those statistics, rather than any individual vector, that serves as our proxy for the model of structure available to speakers.

By keeping the reference frame fixed (the same shallow model across cohorts), we observe that adult geometry returns to an initial, denser configuration, whereas intermediate cohorts show shifts relative to that fixed frame. Our account is that temporal resolution changes with learning (sequence organization becomes more variable and redistributed; results 2.5 and 2.6, and the closing paragraph of 2.2), so the observed geometry moves with respect to a stable baseline. In a system where even relatively stable distributions (e.g., parts of speech and word-boundary patterns) shift over development, a fixed reference frame is essential for detecting and measuring that movement. Thus, the embeddings are approximation tools that make these shifts observable; we do not treat them as children's internal representation spaces.

Further support, such as behavioral data on category confusability, developmental benchmarks, or alignment with known learning trajectories, would be needed to strengthen the causal interpretation.

Response:

We appreciate the reviewer's concern but wish to clarify that our analysis is not framed as a causal account of how individual concepts emerge. Rather, it is a system-level examination of how age-related changes in word-sequence statistics alter the information that supports category discrimination. Whether speakers access this information cannot be determined from their productions alone.

To address this point, we added Section 2.4 examining whether cohort-specific word2vec models help account for children's responses in semantic relatedness and sound-discrimination tasks (OpenNeuro DS000221). Using GAMMs, embedding-based similarity scores improve model fit to trial-level accuracy and response times at ages 5, 7, and 9, relative to a baseline without embeddings. Models trained on child-produced language performed as well as or modestly better than adult-trained alternatives; within these, the 5PLUS model showed the most consistent generalization across ages in this dataset. Taken together, these results suggest that cohort-trained embeddings capture task-relevant regularities that vary with experience. We view this as modest, model-linked support for using embeddings to summarize input structure; it does not imply that embeddings reflect children's cognitive representations or specify a causal mechanism.

In addition, we revised the introduction and discussion to emphasize the limitations of inferences based on surface-level statistics. We now make explicit that: (1) these analyses are based on production data only, (2) similar behavior can reflect different

underlying processes, and (3) cohort-specific interactions between response time and accuracy suggest shifts in attention and strategy that we fold into our interpretation of the representational dynamics. We argue that mapping structural properties of language to cognitive development requires triangulation with behavioral and neurocognitive data, and we see this work as a step in that direction, not as a claim to mental content, but as a proposal for how production structure may constrain the development and expression of mental models over time.

We also make explicit, in Methods and now in Discussion, that developmental comparisons have inherent limits: as temporal resolution changes with learning, age cohorts are not i.i.d. samples from a single process. Treating them as if they were generated by the same random variable would overstate what cross-cohort inference can support. Our approach therefore uses a fixed, shallow reference frame to measure how structure moves over development and reports those movements descriptively. This is a general methodological issue in modeling developmental data in cognitive science, not specific to our study: because the underlying data-generating process seems to shift in time, we focus on changes in structure rather than on causal mechanisms.

Consistent with this perspective, we agree that embedding-based analyses alone do not license strong cognitive interpretations. We have revised the framing of our claims throughout to clarify that our findings reflect system-level regularities in production, not causal mechanisms of learning or representation.

Additionally, showing robustness across different embedding methods or controlling for confounding variables like corpus size and lexical diversity would enhance the argument.

Response:

Thank you for this helpful suggestion. We addressed it in three ways.

1. **Corpus size / lexical diversity.** We constructed near-matched sub-corpora (~1.7M tokens per cohort) by sampling consecutive sequences and excluding collections as needed, so results are not driven by sample size. Lexical diversity naturally increases with age; we treat this as part of the developmental signal, not a nuisance variable, and we quantify its relation to sequence position and part-of-speech in sections 2.5–2.6 and Fig. 4.
2. **Why we do not swap embedding algorithms; within-algorithm sensitivity.** CBOW, skip-gram, GloVe, and BERT implement different error signals / learning objectives (e.g., local vs. global co-occurrence; forward vs. masked-token losses). Because our goal is to trace positional error patterns as content words diversify relative to context, swapping algorithms would mainly surface implementation details of those objectives and risk conflating model-specific artefacts with the developmental signal (cf. Shcherbakova et al., 2025, *Different models, different assumptions, different findings*). Instead, we keep the learner and hyper-parameters fixed across age slices. For transparency, we report within-algorithm sensitivity: we retrained Word2Vec with dimensionalities 10/20/50 and windows 2/5/10; all configurations reproduce the non-linear

developmental curve, with shifts in shape and the inflection (see Supplementary Analyses Fig. A15/16 and benchmarking notebook).

- 3. Robustness via input perturbations and open benchmarking.** Rather than varying the model, we assess robustness by scrambling the input while keeping the learner identical (backward, shuffled, and position-coded sequences) and by providing an OSF “benchmarking sandbox” notebook that allows alternative implementations. These resources are informative for a separate methods paper on how learning objectives interact with distributional dynamics; here, our scope is the data-driven theoretical result linking network-summarized structure to changes in surface productions (reported in sections 2.5–2.6).

- Do you feel that the paper will influence thinking in the field?

If the assumptions underlying the interpretation of embedding space as a proxy for cognitive structure can be substantiated, this approach has the potential to influence thinking in the field. The work presents a promising framework for linking system-level changes in language use to representational development, which may inspire further studies using computational tools to analyze developmental corpora.

However, the current version of the paper would benefit from more theoretical clarity and empirical support before it can be widely adopted as a model of cognitive development.

Response:

Thank you for this thoughtful assessment, and for suggesting that the approach could be broadly useful. We agree that theoretical clarity and explicit empirical links are essential. In the revision we (i) make our scope explicit: this is a system-level, data-driven analysis of how age-related changes in sequence structure alter the information available to learners, not a causal model of cognitive development; (ii) clarify that embeddings are used only as a measurement tool to summarize distributional structure; and (iii) state our theoretical stance plainly: conceptual structure is co-determined within a system of interdependent regularities, with sequential structure serving as an interface for quantifying developmental change, and individual speakers approximating and exploiting this evolving code rather than storing a fixed canonical set of concepts.

We expanded the Discussion to clarify the theoretical framing and scope and to explain how the reported patterns relate to cognitive development. We hope these revisions address the request for clearer framing and empirical grounding.

- Does the paper represent an advance in understanding which may influence thinking in the field?

Yes, the paper represents a conceptual advance by proposing that language learning can be understood through changes in the structure of distributional semantics over time. It encourages researchers to look beyond isolated word learning and instead consider global structural shifts in the linguistic system. However, the influence of this work will depend on the authors' ability to clearly

articulate and justify how their findings map onto cognitive development.

- Does the article present an original study, new analysis, new model, or a direct or extended replication of previous work?

Yes, the study is original in both dataset and methodology. It introduces a new way of modeling and visualizing developmental language change using low-dimensional Word2Vec models trained on age-specific corpora, combined with t-SNE projections, distance metrics, and statistical shape alignment analyses (e.g., Procrustes and Mantel tests).

- Are the data and analysis technically sound? Are they appropriate to answer the research question, e.g., are causal research questions addressed on the basis of causal, rather than correlational evidence?

The analyses are technically competent and methodologically appropriate for a correlational, exploratory study. The authors appropriately use nonparametric and permutation-based methods to examine geometric differences in embedding space. However, the conclusions drawn remain correlational, and the manuscript would benefit from a clearer acknowledgment that no causal inference can be made from the current design.

Response:

We agree with the reviewer and now state more clearly that the findings do not support causal inference. We clarify scope and add targeted analyses, while keeping the paper's identity as a data-driven theoretical contribution. We emphasize that similar surface patterns can arise from different underlying processes. We view this many-to-one (\approx one-in-context) mapping as a feature of the code, not a flaw: linguistic regularities enable coordination while permitting idiosyncratic implementations shaped by affordances and individual differences. In this manuscript we remain at the system-level description; linking implementations to the code requires converging evidence. We sketch how such tests could proceed by aligning model-derived information profiles with changes in neural oscillations and eye movements, alongside behavior.

In Section 2.4, we mention cohort-specific interactions between response time and accuracy, which suggest that changes in representational structure are tied to shifts in attention and task strategy. These results are folded into the discussion as part of a broader point about the need to study how information is distributed over time, and how surface statistics can reflect, but not fully explain, underlying developmental changes. A full treatment is beyond the scope of the present paper.

- Does the paper provide strong evidence for its conclusions?

The evidence is consistent and multifaceted within the scope of the chosen methods, but the conclusions about internal cognitive models are stronger than what the data directly support. The embedding patterns are compelling, but without triangulation from behavioral data or model-independent developmental findings, the cognitive interpretation remains speculative.

- Is the study question important to scientists for a sub-field of psychology?

Yes. Understanding how children organize linguistic knowledge over time is a fundamental question in developmental psychology and psycholinguistics. This study contributes to that discourse by providing a data-driven, systems-level approach to analyzing category structure in child language.

- Are there any special ethical concerns arising from the use of animals or human subjects?

No. The study uses publicly available, de-identified corpora from CHILDES and other sources. There are no direct human subjects or ethical concerns involved.

- Was the study preregistered and if so, did the authors follow the preregistration?

The manuscript does not mention preregistration. While preregistration is not always expected in corpus-based or exploratory modeling studies, some transparency about analytic choices and flexibility would be appreciated.

Response:

Thank you. We agree that transparency about analytic choices and flexibility is important. Because this is a data-driven theory paper in which the model serves as a measuring instrument, we now separate design choices from estimation choices. Design choices (e.g., cohort slicing aligned with developmental phases; consecutive sampling with exclusions to obtain near-matched size and age distributions) define the estimand and are reported up front; they are not tuned to outcomes. Estimation choices are constrained by fixing a single CBOW learner and hyper-parameters across age slices and using one preprocessing pipeline. Where reasonable alternatives exist, we document them, run targeted sensitivity checks (window size, dimensionality) and sequence-integrity controls (backward, position-coded, shuffled), and provide a runnable OSF notebook with all settings and seeds. These steps make the scope and impact of our analytic flexibility explicit without reframing the study as a comparison of model architectures.